# Genetic identification of a common collagen disease in Puerto Ricans via identity-by-descent mapping in a health system

Gillian Morven Belbin[1,2,3], Jacqueline Odgis[2], Elena P Sorokin[4], Muh-Ching Yee[5], Sumita Kohli[1], Benjamin S Glicksberg[2,3,6], Christopher R Gignoux[4], Genevieve L Wojcik[4], Tielman Van Vleck[1], Janina M Jeff[1], Michael Linderman[2,3†], Claudia Schurmann[1†], Douglas Ruderfer[7,8,9‡], Xiaoqiang Cai[2], Amanda Merkelson[1], Anne E Justice[10], Kristin L Young[10], Misa Graff[10], Kari E North[10], Ulrike Peters[11,12], Regina James[13], Lucia Hindorff[14], Ruth Kornreich[2], Lisa Edelmann[2], Omri Gottesman[1§], Eli EA Stahl[1,3,6,7], Judy H Cho[1,2,15], Ruth JF Loos[1,16], Erwin P Bottinger[1#], Girish N Nadkarni[1], Noura S Abul-Husn[1,2,3§], Eimear E Kenny[1,2,3,9]*

*For correspondence:
eimear.kenny@mssm.edu

Present address: †Department of Computer Science, Middlebury College, Middlebury, United States; ‡Departments of Medicine, Psychiatry and Biomedical Informatics, Vanderbilt Genetics Institute, Vanderbilt University Medical Center, Nashville, United States; §Regeneron Pharmaceuticals, Tarrytown, United States; #Berlin Institute of Health, Berlin, Germany

[1]The Charles Bronfman Institute for Personalized Medicine, Icahn School of Medicine at Mount Sinai, New York, United States; [2]Department of Genetics and Genomics, Icahn School of Medicine at Mount Sinai, New York, United States; [3]The Icahn Institute of Genomics and Multiscale Biology, Icahn School of Medicine at Mount Sinai, New York, United States; [4]Department of Genetics, Stanford University School of Medicine, Stanford, United States; [5]Department of Plant Biology, Carnegie Institution for Science, Stanford, United States; [6]Harris Center for Precision Wellness, Icahn School of Medicine at Mt Sinai, New York, United States; [7]Broad Institute, Cambridge, United States; [8]Division of Psychiatric Genomics, Icahn School of Medicine at Mt Sinai, New York, United States; [9]Center for Statistical Genetics, Icahn School of Medicine at Mt Sinai, New York, United States; [10]Department of Epidemiology, University of North Carolina at Chapel Hill, Chapel Hill, United States; [11]Public Health Sciences Division, Fred Hutchinson Cancer Research Center, Seattle, United States; [12]Department of Epidemiology, University of Washington School of Public Health, Seattle, United States; [13]National Institute on Minority Health and Health Disparities, National Institutes of Health, Bethesda, United States; [14]National Human Genome Research Institute, National Institutes of Health, Bethesda, United States; [15]Division of Gastroenterology, Icahn School of Medicine at Mount Sinai, New York, United States; [16]The Mindich Child Health and Development Institute, Icahn School of Medicine at Mount Sinai, New York, United States

**Abstract** Achieving confidence in the causality of a disease locus is a complex task that often requires supporting data from both statistical genetics and clinical genomics. Here we describe a combined approach to identify and characterize a genetic disorder that leverages distantly related patients in a health system and population-scale mapping. We utilize genomic data to uncover components of distant pedigrees, in the absence of recorded pedigree information, in the multi-ethnic BioMe biobank in New York City. By linking to medical records, we discover a locus associated with both elevated genetic relatedness and extreme short stature. We link the gene,

*COL27A1*, with a little-known genetic disease, previously thought to be rare and recessive. We demonstrate that disease manifests in both heterozygotes and homozygotes, indicating a common collagen disorder impacting up to 2% of individuals of Puerto Rican ancestry, leading to a better understanding of the continuum of complex and Mendelian disease.

DOI: https://doi.org/10.7554/eLife.25060.001

## Introduction

During the past two decades major advances in deciphering the genetic basis of human disease have resulted in thousands of disorders that are now understood at a genetic level (*Hamosh et al., 2000*; *McCarthy et al., 2013*). This progress has led to the integration of genomic sequencing in clinical care, especially for the diagnosis of rare genetic disease (*Bamshad et al., 2011*; *Manolio et al., 2013*), and clinical sequencing is increasingly offered to patients with known or suspected genetic disorders. In the past few years, large national and international efforts (*Manolio et al., 2015*; *Philippakis et al., 2015*; *Chong et al., 2015a*) have emerged to enable patients and health systems to share knowledge of rare genetic disorders and improve genetic testing, resulting in improved healthcare management and outcomes for patients. In parallel, many large regional and national biobank efforts (*Collins and Varmus, 2015*; *Ashley, 2015*; *Collins, 2012*) are underway to enable the broad integration of genomics in health systems for genetic identification of disease (*Dewey et al., 2016*). Such efforts have recently revealed clinically actionable variants (*Dewey et al., 2016*; *Feldman, 2016*) and genetic disorders segregating at higher frequencies in general patient populations than previously suspected. The increased promulgation of genomics in health systems represents an opportunity to improve diagnostic sensitivity for more precise therapeutic intervention and better health outcomes (*Ashley, 2016*).

Despite this progress, most genetic diseases are still under-diagnosed (*Abul-Husn et al., 2016*) or misdiagnosed (*Yang et al., 2014*; *Taylor et al., 2015*; *Centers for Mendelian Genomics et al., 2015b*). A number of barriers exist for wholesale genetic testing and diagnoses, including incomplete standardized guidelines for interpreting genetic evidence of disease (*Amendola et al., 2016*), variable penetrance or expressivity of phenotype (*Katsanis, 2016*), and that the causal variant may be missed or mis-assigned during testing (*Manrai et al., 2016*). The latter is a particularly pernicious problem in non-European populations due to systematic biases in large genomic and clinical databases (*Popejoy and Fullerton, 2016*; *Petrovski and Goldstein, 2016*). These challenges have led several research groups to attempt to genetically identify disease by examining patient health patterns using data from the Electronic Health Record (EHR) (*Gottesman et al., 2013*; *Green et al., 2016*). EHRs contain comprehensive information on medical care throughout a patient's life, including medications, medical billing codes, physician notes and generated reports (i.e. pathologic, genetic and radiologic reports). EHRs have been used to clinically characterize well-known genetic disorders, but have been of limited success for the vast cadre of less-characterized or unknown disorders (*Blair et al., 2013*).

The gold standard of genetic disorder diagnosis involves testing both patient and family members to confirm Mendelian segregation of the suspected underlying pathogenic variant (*ClinGen et al., 2015*; *ClinGen Resource et al., 2015*; *Gahl et al., 2016*). However, as genomic data becomes more ubiquitous in health systems, it can be used to detect genetic relationships in the absence of known family and pedigree information. Specifically, components of pedigrees can be uncovered within the general population; particularly those that have experienced recent founder effects. Pairs of individuals who are related share genetic homology in the form of long genomic haplotypes. These haplotypes are considered to be identical-by-descent (IBD) if they are inherited from a common ancestor without any intervening recombination. The chance of any two people sharing a tract of their genome IBD decays exponentially, with a ~50% reduction in the chance of sharing per generation. However, when IBD sharing does occur, the length of an IBD segment can remain long even between distantly related individuals. In practice, long tracts of IBD (>3 cM) can be accurately detected using genetic data between individuals with a common ancestor from the past 4–50 generations (*Browning and Browning, 2012*). Detection of IBD haplotypes can allow for the identification of distantly related patients with a genetic disorder driven by a locus inherited from a founding ancestor who brought the disease mutation into a population (*Houwen et al., 1994*;

**eLife digest** Diseases often run in families. These disease are frequently linked to changes in DNA that are passed down through generations. Close family members may share these disease-causing mutations; so may distant relatives who inherited the same mutation from a common ancestor long ago. Geneticists use a method called linkage mapping to trace a disease found in multiple members of a family over generations to genetic changes in a shared ancestor. This allows scientists to pinpoint the exact place in the genome the disease-causing mutation occurred. Using computer algorithms, scientists can apply the same technique to identify mutations that distant relatives inherited from a common ancestor.

Belbin et al. used this computational technique to identify a mutation that may cause unusually short stature or bone and joint problems in up to 2% of people of Puerto Rican descent. In the experiments, the genomes of about 32,000 New Yorkers who have volunteered to participate in the BioMe Biobank and their health records were used to search for genetic changes linked to extremely short stature. The search revealed that people who inherited two copies of this mutation from their parents were likely to be extremely short or to have bone and joint problems. People who inherited one copy had an increased likelihood of joint or bone problems.

This mutation affects a gene responsible for making a form of protein called collagen that is important for bone growth. The analysis suggests the mutation first arose in a Native American ancestor living in Puerto Rico around the time that European colonization began. The mutation had previously been linked to a disorder called Steel syndrome that was thought to be rare. Belbin et al. showed this condition is actually fairly common in people whose ancestors recently came from Puerto Rico, but may often go undiagnosed by their physicians. The experiments emphasize the importance of including diverse populations in genetic studies, as studies of people of predominantly European descent would likely have missed the link between this disease and mutation.

DOI: https://doi.org/10.7554/eLife.25060.002

*Kenny et al., 2009*; *Henden et al., 2016*; *Meta-Analysis of Glucose and Insulin-related traits Consortium (MAGIC) et al., 2010*; *Traherne et al., 2016*; *Shaw et al., 2015*; *Ko et al., 2014*; *Lalli et al., 2014*). This is the principle underlying population-scale disease mapping approaches that combine IBD sharing and statistical association to discover novel disease loci, so called IBD-mapping.

By detecting genetic relatedness, as inferred by IBD sharing, we hypothesized that we may be able to detect hereditary forms of disease in an EHR-linked biobank. With over 38000 participants, the BioMe biobank, at the Icahn School of Medicine at Mount Sinai, New York City (NYC), is one of the most diverse cohorts ascertained at a single urban medical center under a uniform study protocol. Participants are largely from the local Upper East Side, Harlem and Bronx communities, and represent broad ancestral, ethnic, cultural, and socioeconomic diversity. We initially focused on adult height, which is easily measurable, stable over the adult life course, and one of the most abundantly recorded clinical parameters in EHRs. Height is known to be highly heritable and polygenic (*Visscher et al., 2010*; *Lango Allen et al., 2010*), however, extremes of short stature can be caused by rare variants in single genes with large effect sizes (*Durand and Rappold, 2013a*). Although, many genetic syndromes are known to cause short stature, most of the time no definitive etiology underlying short stature is found in patients. Here we used loci associated with elevated genetic relatedness as measured by IBD to map a locus underlying extreme short stature in the BioMe biobank, and linked it to a known, but little characterized, collagen disorder previously thought to be rare. By interrogating a large global diversity panel, we demonstrated that this variant is actually common in Puerto Rican populations. Furthermore, we leveraged the EHR to show significant musculoskeletal disease in both heterozygous and homozygous patients, indicating the disease is not simply a recessive disorder as had previously been thought. Finally, we showed how this work can generate broad insights for sustainable adoption and large-scale dissemination of genomic medicine.

## Results

### Detecting patterns of diversity, founder effects and relatedness in the BioMe biobank from New York City

The BioMe biobank comprises a highly diverse cohort, with over 65% of participants self-reporting as Black/African-American or Hispanic/Latino/a, and over 35% born outside mainland US, representing more than 110 countries of origin. First we estimated patterns of direct relatedness in a subset of BioMe participants genotyped on the Illumina OmniExpress array (N = 11212) by detecting pairwise identity-by-state using RELATEAdmix (*Moltke and Albrechtsen, 2014*), a method that accounts for admixture in populations (*Figure 1—figure supplement 1*). We observed that 701 individuals had primary (parent-child, sibling) or secondary (avuncular, grandparental) relationship with another participant in BioMe, and we removed these individuals from all downstream analysis. Next we devised a strategy to divide the diverse BioMe biobank into population groups for downstream analysis. We combined genotype data for BioMe participants (N = 10511) with 26 global populations from the 1000 Genomes Project (1KGP; N = 2504) (*The 1000 Genomes Project Consortium, 2015*) and two additional panels of Native American (N = 43) (*Mao et al., 2007*) and Ashkenazi Jewish populations (N = 100) (see Methods). Using a common set of 174468 SNPS we performed principal component analysis (*Price et al., 2006*) (PCA; *Figure 1—figure supplement 2*). Based on both self-reporting and patterns of genetic diversity observed in BioMe participants, we stratified individuals into four broad population groups. The first group self-reported as European American, but were also genetically identified as Ashkenazi Jewish (AJ; N = 808) as they clustered distinctly with an AJ reference panel and separately from other European ancestry groups in PCA space (*Figure 1—figure supplement 3*). The other three groups we defined using self-reported race/ethnicity categories, African-American (AA; N = 3080), Hispanic/Latino/a (H/L; N = 5102) and European-Americans with no AJ genetic ancestry (Non-AJ EA; N = 1270) (*Figure 1—figure supplement 3*). An additional 251 individuals who reported 'Mixed' (N = 89) or 'Other' (N = 162) ethnicity were excluded from further analysis.

To evaluate signatures of distant relatedness BioMe biobank participants, we estimated sharing of genomic tracts IBD >3 cM between every pair of individuals using the GERMLINE software (*Gusev et al., 2009*). The minimum length of 3 cM was chosen based on reports of elevated type I error in call rates of smaller lengths (*Gusev et al., 2012*; *Chiang et al., 2016*). It is known that population-level rates of distant relatedness are observed to be particularly elevated after population bottlenecks (*i.e.* in founder populations) (*Browning and Thompson, 2012*). We summed the length of all IBD-tracts shared between a given pair of individuals if they shared more than one tract and examined the distribution of pairwise sharing at a population level. We observed elevated levels of distant relatedness in both the AJ (median summed length of IBD sharing within population = 44.7 cM; 95% C.I. = 44.66–44.82 cM) and HL (16.2 cM; 16.18–16.22 cM) populations, compared to AA (3.77 cM; 3.76–3.77 cM) or non-AJ EA (4.5 cM; 4.45–4.55 cM) populations (*Figure 1A*). This is congruent with previous reports of founder effects in both AJ populations (*Need et al., 2009*) and in some H/L populations (*Moreno-Estrada et al., 2013*).

Hispanic or Latino/a is a broad ethnic label encompassing myriad populations with origins in Northern, Southern or Central America, century-long roots in New York City, and genetic ancestry from Africa, Europe and the Americas. To explore the signature of a founder effect in the BioMe H/L population, we leveraged self-reported and genetic information about sub-continental ancestry. By self-reporting, the H/L participants in BioMe were born in New York City (NYC) (40%), Puerto Rico (24%), Dominican Republic (19%), Central/South America (12%), Mexico (2%) or other Caribbean Island (2%) (*Figure 1B*). We examined IBD tract length distributions within H/L sub-continental populations and observed that the founder effect was predominantly driven in the Puerto Rican-born group (*Figure 1C*). We assembled a cohort of Puerto Ricans including BioMe participants who were either born in Puerto Rico or, were born in NYC and had 2 parents or 3–4 grand parents who were born in Puerto Rico (N = 1245). Approximately 5086 NYC-born H/L individuals did not have recorded parental or grandparental country-of-origin, therefore we also devised a selection strategy using PCA analysis. We identified BioMe H/L participants on the cline between the African and European reference panels in PCA space coincident with Puerto Rican-born individuals. We excluded those on the same cline with ancestry from the Dominican Republic or another Caribbean Island (*Figure 1—figure supplement 4*), and counted the remainder (N = 1571) in the Puerto Rican group. In

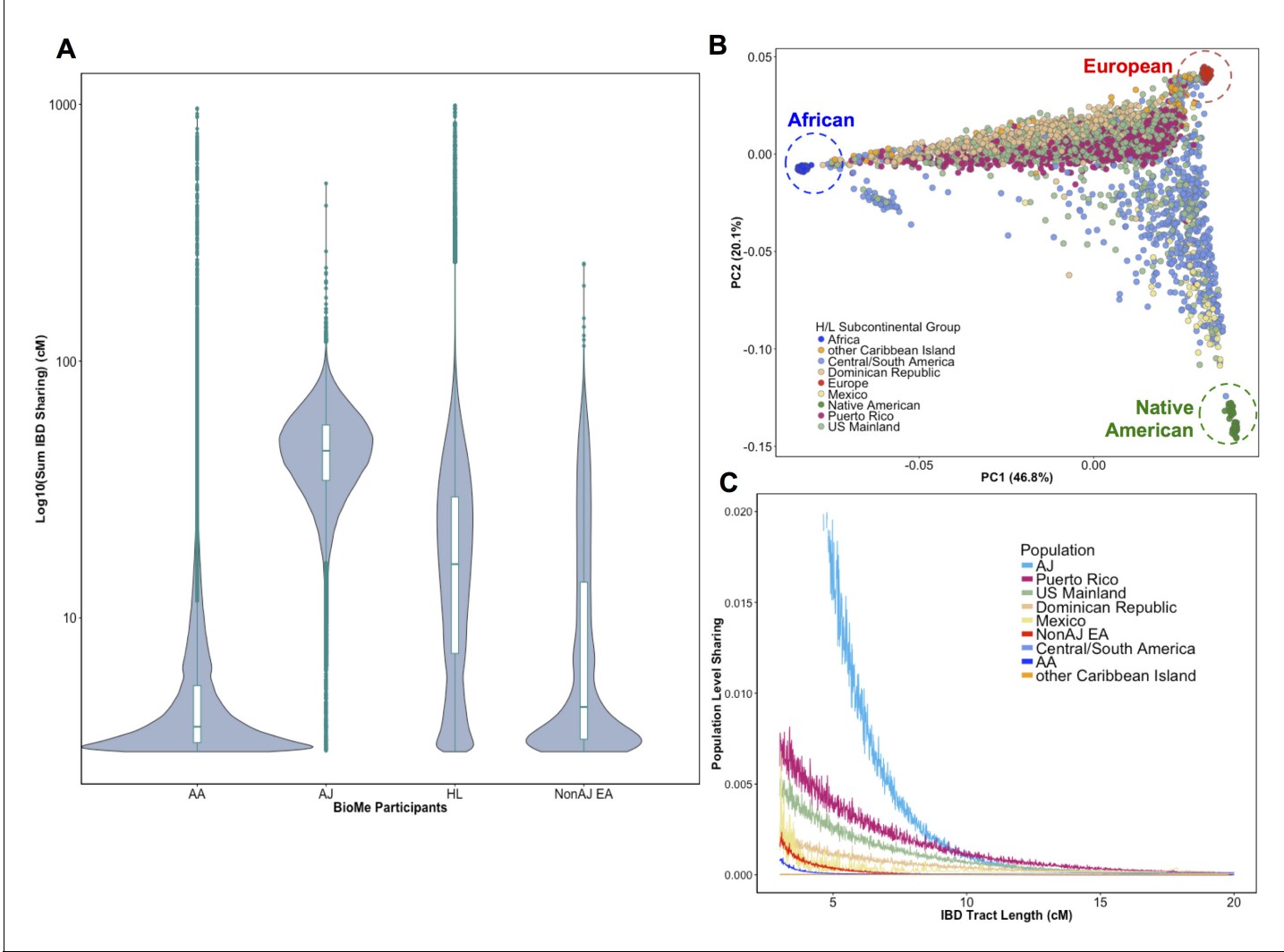

**Figure 1.** Distribution of Identity-by-Descent sharing among Bio*Me* populations. (**A**) Distribution of the pairwise sum of Identity-by-Descent (IBD) sharing (cM) between four broadly defined Bio*Me* populations, namely; African American (AA), Ashkenazi Jewish (AJ), Hispanic/Latino/a (H/L) and Non-Jewish European American (Non-AJ EA). (**B**) Sub-continental diversity in self-reported H/L participants in Bio*Me*. Afro-Caribbean participants fall between European (red) and African (blue) continental reference panels, Mexican and Central/South American H/L participants fall between European and Native American (green) reference panels, mainland US-born participants reside on either cline. (**C**) The tract length distribution of IBD sharing among Bio*Me* populations, normalized by population size. The y-axis represents the proportion population-level sharing ($x / (N*(N-1)/2)$), where $x$ is the sum of the number of pairwise shared IBD tracts and N is the number of individuals per population. The AJ population exhibits the highest level of population level sharing, followed by Puerto Rican born H/L.

DOI: https://doi.org/10.7554/eLife.25060.003

The following figure supplements are available for figure 1:

**Figure supplement 1.** Identity-by-State analysis to test for presence of relatedness in four Bio*Me* populations.
DOI: https://doi.org/10.7554/eLife.25060.004

**Figure supplement 2.** Principal Component Analysis of all Bio*Me* participants show extensive global genetic diversity.
DOI: https://doi.org/10.7554/eLife.25060.005

**Figure supplement 3.** Principal Component Analysis in Bio*Me* AA and EA reveals sub-continental structure.
DOI: https://doi.org/10.7554/eLife.25060.006

**Figure supplement 4.** Inference of suspected Puerto Rican ancestry individuals *via* principal component analysis.
DOI: https://doi.org/10.7554/eLife.25060.007

total, we estimated 2816 H/L in the BioMe discovery cohort were of Puerto Rican ancestry, and focused the downstream analysis on this group as the largest founder population in BioMe.

## Detecting a locus shared identical-by-descent underlying extreme short stature in Puerto Ricans

Next we tested the hypothesis that rare, recessive disease variants may have arisen to appreciable frequency in the Puerto Rican founder population. We linked genomic data to clinical data in the Electronic Health Record (EHR) of the Mount Sinai Health System. We focused on height, a stable and ubiquitous health measure. Clinically, rare instances of growth failure or 'short stature' may be caused by a large heterogeneous group of genetic disorders (i.e. skeletal dysplasias) (*Durand and Rappold, 2013b*). We first extracted measures of height for the Puerto Rican adult population of BioMe (mean age = 55.3, standard deviation (s.d.)=16.1). After making exclusions based on age (>=18 years old for women, >=22 years old for men, and <80 years old in both sexes, 194 individuals in total), mean height measurements (mean height = 5' 8.2', s.d. = 3.2' for men; mean height = 5' 2.8' s.d. = 2.8' for women) were consistent with those reported for Puerto Rican populations in a recent global study on height (*NCD Risk Factor Collaboration (NCD-RisC), 2016*). We noted that 56 Puerto Ricans met the clinical definition of short stature (*Cohen et al., 2008*) (range of short stature 5'1'−4'0' in men, and 4'8'−3'8' in women) defined as 2 standard deviations below the population-specific mean for men and women separately (*Figure 2—figure supplement 1*).

To test for recently arisen, recessive variants underlying clinical short stature in Puerto Ricans, we implemented a previously published pipeline for 'IBD mapping' (*Kenny et al., 2009*; *Vacic et al., 2014*) (*Figure 2—figure supplement 2*). We first clustered participants into 'cliques' of 3 or more individuals whom, at a given genomic region, shared overlapping homologous IBD tracts of at least 0.5 cM in length in either a heterozygous or homozygous state. Membership in a clique indicates the sharing of a recent common ancestor at that locus, from which the homologous IBD tract was jointly inherited. Clustering of IBD into cliques in the Puerto Rican population (N = 2816) yielded 1434421 IBD-cliques after quality control filters (see Methods). The site frequency spectrum of IBD-cliques (*Figure 2—figure supplement 3*) demonstrates an expected exponential distribution of clique sizes (of 3–77 haplotypes), representing a class of rare IBD haplotypic alleles (allelic frequency 0.0005–0.0137). To test whether any cliques of IBD haplotypes were significantly associated with height we performed genome-wide association of height as a continuous trait under a recessive model using PLINKv1.9 (*Purcell et al., 2007*; *Chang et al., 2015*), including the first five PCA eigenvectors as covariates (see Materials and methods). We restricted analysis to 480 out of 1434421 cliques that contained at least 3 individuals who were homozygous for the shared haplotype. Adjusting for 480 tests (Bonferroni adjusted threshold $p<1\times10^{-4}$) one IBD-clique achieved a genome-wide significant signal at the locus 9q32 (IBD-clique frequency = 0.012; β = −3.78; $p<2.57\times10^{-11}$) (*Figure 2A*), spanning a large mapping interval chr9:112 MB-120MB. The clique contains 59 individuals, 56 of whom are heterozygous and 3 are homozygous for the associated IBD haplotype.

## Fine-mapping short stature locus reveals putative link to mendelian syndrome

The three individuals driving the recessive signal, two women and one man, were less than 2.5 s.d. shorter (height reduction range 6'−10') than the population mean for height in the Puerto Rican cohort (*Figure 2B*). The IBD-haplotypes driving the signal spanned a genic region with several candidate loci, and the minimum shared boundary overlapped a single gene, *COL27A1*, which encodes for Collagen Type XXVII, Alpha 1 (*Figure 2C*). We performed whole genome sequencing (WGS) of the three homozygous individuals, and an additional short-statured individual that we observed to possess a homozygous IBD haplotype that was both directly upstream of and highly correlated with the top IBD-clique. Individuals were sequenced to a depth of 4-18X coverage (*Supplementary file 1*). Examination of variants that were observed in at least 6 copies between the four individuals (to account for sequencing error or missing calls) revealed a single candidate coding allele, a missense mutation in Collagen Type XXVII, Alpha 1 (*COL27A1*, g.9:116958257.C>G, NM_032888.1, p.G697R, rs140950220) (*Supplemental file 2*). In silico analysis suggest that this glycine residue is highly conserved, and that a molecular alteration to arginine at this position is predicted to be damaging (SIFT score = 0.0; PhyloP score = 2.673; GERP NR score = 5.67). These findings are consistent with a

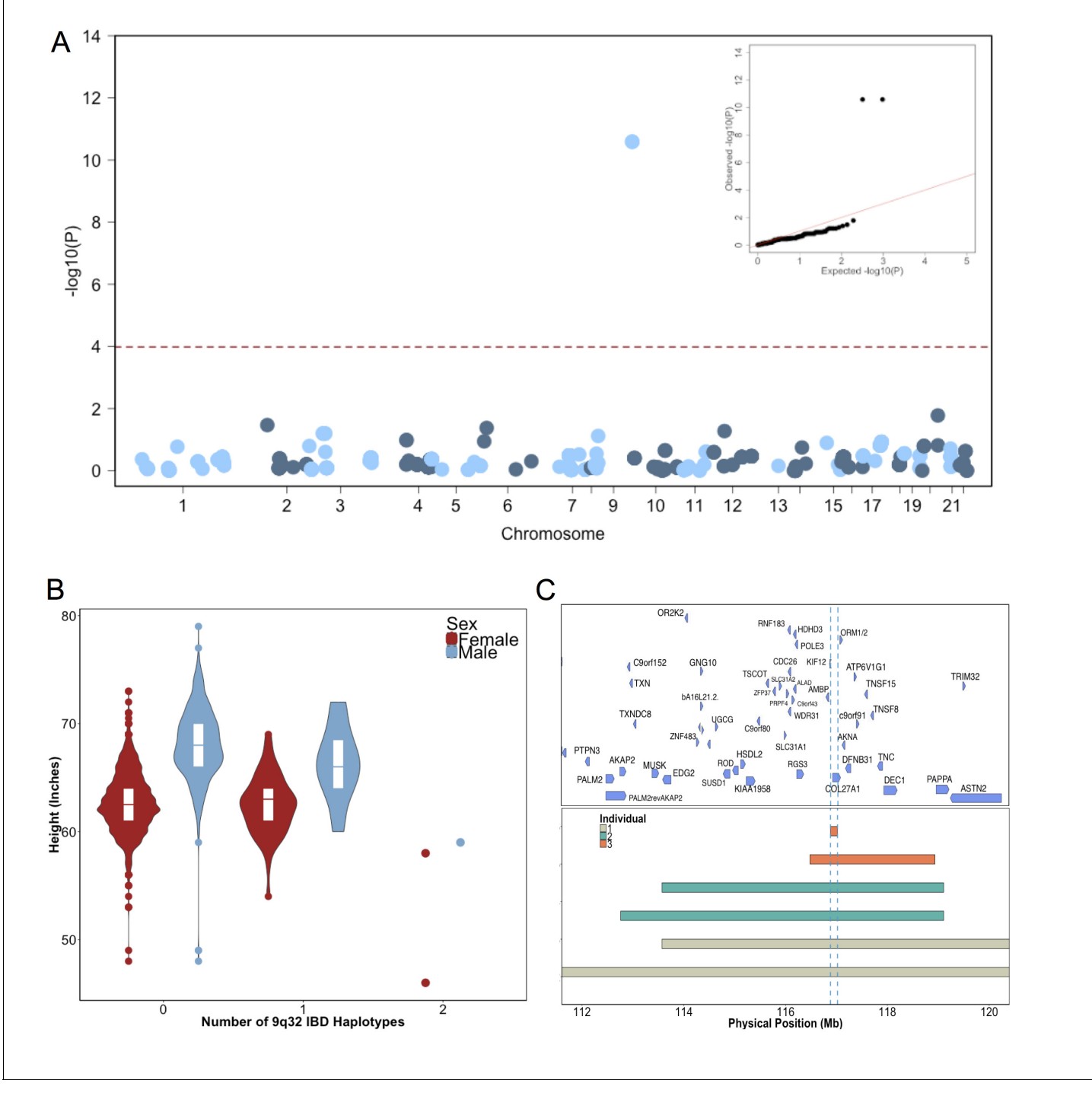

**Figure 2.** Identity-by-Descent mapping reveals locus associated with short stature. (**A**) Identity-By-Descent (IBD) mapping for height in Bio*Me* Puerto Ricans using a recessive model. Analysis was restricted to IBD-cliques where at least three individuals were homozygous. Only one IBD-clique achieved Bonferroni significance (at 9q32). (**B**) Distribution of height among Puerto Rican individuals who carry either 0,1 or 2 copies of the IBD-haplotype reveals a large recessive effect. Homozygous individuals (those carrying 2 copies of the IBD-haplotype) are on average 6–10′ shorter than the population mean for Puerto Rican ancestry individuals. (**C**) The minimum shared boundary of the significant IBD-haplotype between the three homozygous individuals (represented by the dashed blue line). The top panel depicts known genes at the 9q32 locus. The minimum shared boundary of the IBD overlaps the gene *COL27A1*..

DOI: https://doi.org/10.7554/eLife.25060.008

The following figure supplements are available for figure 2:

**Figure supplement 1.** Distribution of height in the Bio*Me* Puerto Ricans stratified by sex.

*Figure 2 continued on next page*

*Figure 2 continued*

DOI: https://doi.org/10.7554/eLife.25060.009

**Figure supplement 2.** A framework for biomedical discovery *via* Identity-by-Descent (IBD) mapping.

DOI: https://doi.org/10.7554/eLife.25060.010

**Figure supplement 3.** Site Frequency Spectrum of IBD-cliques in Bio*Me* Puerto Ricans follows an exponential distribution.

DOI: https://doi.org/10.7554/eLife.25060.011

**Figure supplement 4.** Genome Wide Association of Height in N = 2622 Puerto Ricans under a recessive model appears to be well-calibrated but reveals no genome-wide significant signal.

DOI: https://doi.org/10.7554/eLife.25060.012

**Figure supplement 5.** A structural model of glycine-to-arginine substitution within a collagen triple helix.

DOI: https://doi.org/10.7554/eLife.25060.013

**Figure supplement 6.** Depth of IBD-sharing across the genome in Bio*Me*.

DOI: https://doi.org/10.7554/eLife.25060.014

**Figure supplement 7.** Size of Homologous IBD-Clique Membership Across the Genome in Bio*Me*.

DOI: https://doi.org/10.7554/eLife.25060.015

recent report implicating the same *COL27A1* variant as causal for the rare orthopedic condition Steel syndrome in a Puerto Rican family (*Gonzaga-Jauregui et al., 2015*). First described in 1993, the main clinical features of Steel syndrome include short stature, bilateral hip and radial head dislocations, carpal coalition (fusion of the carpal bones), scoliosis, *pes cavus* (high arches), and dysmorphic features (*Steel et al., 1993*).

To confirm the link between the IBD haplotype and the putative causal variant, we calculated the concordance between the IBD haplotype and carrier status of the *COL27A1*.pG697R variant by genotyping all of the homozygotes and carriers of the top IBD-clique in the recessive model (N = 59), along with a panel of age- and sex-matched controls (N = 59). This demonstrated 100% concordance between the *COL27A1*.pG697R variant and the significant IBD-haplotype in homozygotes (*Supplementary file 3*). We note that two Puerto Rican participants in the phase 3 1KGP reference panel were carriers of the *COL27A1*.pG697R variant, raising the possibility that we may have been able to detect this association using more a traditional SNP association approach. Therefore, we performed genome-wide association in the same Puerto Rican cohort (N = 2622) by first imputing the 1KGP panel and re-running the recessive test as described above (n = 10007795 imputed and genotyped SNPs with an INFO score of >0.3 and at least two observations of homozygotes). The recessive model appeared to be well calibrated ($\lambda$ = 1.02), however, we observed no genome-wide significant signal (*Figure 2—figure supplement 4*). Association with the *COL27A1*.pG697R variant was the 11775th most significant association (MAF = 0.014: $\beta$ = $-3.0$; p<0.001). Upon examination of the correlation between the imputed *COL27A1*.pG697R and the true carrier status of homozygotes, we noted a concordance of only 66.67%, indicating that the IBD haplotype was a better tag of the true *COL27A1*.pG697R homozygous state compared to 1KGP imputation in the Puerto Rican cohort (*Supplemental file 3*).

The association between *COL27A1*.pG697R and clinical short stature was replicated using a cohort of 1775 individuals of self-reported Puerto Rican ancestry from Bio*Me* that were not included in the discovery analysis, that were genotyped on the Illumina Infinium Multi-Ethnic Genotype Array (MEGA) as part of the Polygeneic Architecture using Genomics and Epidemiology (PAGE) Study. The *COL27A1*.pG697R (rs140950220) variant was directly genotyped on MEGA, and an association of the variant under a recessive model resulted in a strong signal of association (allele frequency = 0.017; $\beta$ = $-3.5$; s.e. = 0.70; p<$4.87 \times 10^{-07}$). The replication analysis revealed 51 additional Bio*Me* carriers and two individuals that were homozygous for the variant. Both carrier and affected status was confirmed *via* independent genotyping and Sanger sequencing (see Materials and methods). The two homozygous participants were both short statured (2.4 and 3.6 s.d from the sex specific population mean).

## Evidence from electronic health records supports suspected cases of steel syndrome

To determine whether there was any clinical evidence to validate the link between the *COL27A1*.pG697R variant and Steel syndrome, a clinical expert manually reviewed the electronic health

records (EHR), including clinical diagnoses, surgical procedures, and radiology reports of the five participants (3 women, 2 men, age range 34–74 years) homozygous for the *COL27A1*.pG697R variant that we had identified through the discovery and replication efforts. Of note, there was no evidence that any of the five patients had a clinical diagnosis of Steel syndrome. In all five individuals, however, we found EHR-documented evidence of several previously described Steel syndrome characteristics, including developmental dysplasia of the hip (or congenital hip dysplasia), carpal coalition, scoliosis, and cervical spine anomalies (*Table 1*) (*Steel et al., 1993*; *Flynn et al., 2010*). The incidence of cervical spine anomalies, including cord compression and spine surgeries, was higher

**Table 1.** Clinical characteristics of five BioMe participants homozygote, thirty-four carriers and thirty-one non-carriers of the *COL27A1*.pG697R variant using evidence documented in Electronic Health Records (including billing and procedural codes, laboratory, radiologic, and progress notes) compared to features previously reported in *Flynn et al. (2010)*, and *Steel et al., 1993* for 27 children and 7 adults with Steel syndrome.

| Characteristics | EHR-Documented evidence | | Literature Homozygous children | Literature Homozygous Adults | BioMe Homozygous Adults | BioMe Heterozygous Adults <55yoa | BioMe Controls Adults <55yoa |
|---|---|---|---|---|---|---|---|
| Sample size (Gender) | N (N of females) | | 27 (9) | 7(6) | 5 (3) | 34(20) | 31(23) |
| Age | Mean age (years) | | 12.8 | 35.4 | 51.6 | 41.8 | 40.6 |
| Described Characteristics | EHR-Documented Medical History | | N(%) | N(%) | N(%) | N(%) | N(%) |
| Short Stature | Height >= 2 s.d. from pop mean | | 27 (100) | 7 (100) | 5 (100) | 0 (0) | 0 (0) |
| Bilateral hip dislocation | Congenital hip dislocation | | | 4 | 0 | 0 | |
| | Leg length discrepancy | | | 1 | 0 | 0 | |
| | | Total | 27(100) | 7(100) | 5 (100) | 0 (0) | 0 (0) |
| Radial head dislocation | Elbow contractures | | 24 (89) | 7 (100) | 1 (20) | 0 (0) | 0 (0) |
| Carpal coalition | Wrist deformity | | | 1 | 2 | 1 | |
| | Lunotriquetial fusion | | | 1 | 0 | 0 | |
| | Carpel tunnel | | | 0 | 3 | 3 | |
| | | Total | 24 (89) | 6 (86) | 2 (40) | 5(15) | 4(13) |
| Scoliosis | Scoliosis | | 12 (44) | 6 (86) | 2 (40) | 8 (24) | 4 (13) |
| Pes cavus | Pes cavus | | 12 (44) | 0 (0) | 0 (0) | 0 (0) | 0 (0) |
| Cervical spine anomalies | Cervical stenosis | | | 3 | 5 | 0 | |
| | Cervical discitis | | | 1 | 0 | 0 | |
| | Cervical spondylosis | | | 2 | 7 | 3 | |
| | Cervical cord compression | | | 3 | 3 | 1 | |
| | | Total | 3 (9) | 0 (0) | 4 (80) | 7 (21) | 3 (10) |
| Other Characteristics | EHR-Documented Medical History | | | N(%) | N(%) | N(%) | |
| Other spine anomalies | Lumbar spine | | - | - | 1 (20) | 10 (29) | 5 (16) |
| | Thoracic spine | | - | - | 1 (20) | 5(15) | 4(13) |
| Other skeletal disease | Osteoporosis or osteopenia | | - | - | 3 (60) | 3(9) | 1 (3) |
| | Arthritis or degenerative changes | | - | - | 3 (60) | 13(38) | 6 (19) |
| Relevant surgical Interventions | Hip replacement | | - | - | 3 (60) | 0 (0) | 0 (0) |
| | Knee replacement | | - | - | 2 (40) | 0 (0) | 0 (0) |
| | Cervical spine | | - | - | 3 (60) | 2 (6) | 1 (3) |
| | Lumbar spine | | - | - | 1 (20) | 1 (3) | 0 (0) |
| | Thoracic spine | | - | - | 2 (40) | 1 (3) | 1 (3) |

DOI: https://doi.org/10.7554/eLife.25060.016

than previously reported (four out of five patients). There was also evidence of other significant musculoskeletal complications, including lumbar and thoracic spine anomalies in three patients, knee replacements in two patients (both under age 50), and joint degeneration or arthritis in four patients. Together, these data help further our understanding of Steel syndrome-associated characteristics and potential complications that can occur later in life.

## Functional Investigation of *COL27A1.pG697R*

To understand the biological mechanism underlying Steel syndrome, we investigated the functional role of the *COL27A1* gene. *COL27A1* is a fibrillar collagen, which are a class of collagens that contribute to the structural integrity of the extracellular matrix (*Pace et al., 2003*). Enrichment of *COL27A1* RNA expression in vertebrae, as well as long bones, eyes, and lungs has previously been observed in embryonic mice (*Pace et al., 2003*). A mouse deletion of 87 amino acids of the *COL27A1* homolog exhibited severe chondroplasia consistent with clinical features observed in homozygotes (*Plumb et al., 2011*), a similar musculoskeletal phenotype was observed in knockdown of the *col27a1a* and *col27a1b* genes in zebrafish (*Christiansen et al., 2009*). Type alpha-1 collagen genes, of which *COL27A1* is a member, contain a conserved Gly-Xaa-Yaa repeat in their triple helical domain (*Persikov et al., 2004*). Therefore, we hypothesized that the *COL27A1*.pG697R variant may similarly disrupt stability of the *COL27A1* triple helix.

To test this hypothesis we modeled the effect of a glycine-to-arginine substitution in the structure of a prototypical collagen peptide (*Bella et al., 1994*). We observed that the glycine residues occupied the center of the crowded triple helix, and that substitution for a bulkier arginine would likely destabilize helix formation through steric hindrance (*Figure 2—figure supplement 5*). These data provide support for a functional model of the pathogenicity of *COL27A1*.G697R through destabilization of the triple helix, which may occur within developing spinal chords, long bones, and other tissues, resulting in the observed clinical features in homozygotes. We note that many other collagen disorders, including Ehlers-Danlos syndrome (*McGrory et al., 1996*; *Tromp et al., 1995*; *Anderson et al., 1997*), Alport syndrome (*Knebelmann et al., 1992*; *Zhou et al., 1992*) and Osteogenesis Imperfecta (*Starman et al., 1989*; *Shapiro et al., 1992*), are driven by molecular alterations of a glycine in the triple helix of the underlying collagen genes. However, all of these disorders are inherited under an autosomal dominant mode, in contrast to Steel syndrome, which has only been reported as a recessive disease. This analysis raises the question of whether some and/or milder clinical features of Steel syndrome may be present in carriers.

## Assessing the health records of COL27A1.pG697R carriers reveals evidence of musculoskeletal disease

To test for clinical features of Steel syndrome in *COL27A1*.pG697R carriers, we performed two analyses using EHR data. The first was a test for associated medical billing codes (ICD9s) with *COL27A1*.pG697R carrier status, or Phenome-Wide Association Study (PheWAS) (*Denny et al., 2010*; *Denny et al., 2013*). PheWAS analysis is often performed using a general linear model (GLM), however standard implementations often do not account for scenarios where there is a large imbalance between per-test number of cases and controls, rare variants/ICD9s or the presence of elevated distant relatedness. Therefore, in addition to the GLM, we also ran three other score based tests; (i) that use saddlepoint approximation (SPATest) (*Dey et al., 2017*) to account for case:control imbalance; (ii) a linear mixed model (GCTA) (*Yang et al., 2011*) to account for distant relatedness; and (iii) a test that incorporates a bias-reduction for small numbers of observations (Firth test) (*Wang, 2014*). Each test was run using ICD9 codes in all individuals of Puerto Rican ancestry (N = 106 *COL27A1*.pG697R carriers and N = 4480 non-carriers). The five homozygotes were excluded. The ICD9 code was set as the outcome variable and *COL27A1*.pG697R as the primary predictor variable, including age, sex and the first five PCAs as covariates in all tests. To avoid spurious associations, we restricted the analysis to diagnosis codes with at least 3 observations (n = 367 ICD9 codes) amongst carriers.

Results of the GLM test are shown in *Figure 3* and *Table 2*. Of the five significantly associated ICD9 codes (False Discovery Rate (FDR) < 0.05), three involved the musculoskeletal system 730.08 ($p_{GLM} < 7.1 \times 10^{-6}$; odds ratio (OR) = 34.5; 95% Confidence Interval (CI) = 7.4–162), 721.0 ($p_{GLM} < 6.6 \times 10^{-5}$; OR = 5.4; CI = 2.4–12.3), and 716.98 ($p_{GLM} < 4.4 \times 10^{-4}$; OR = 5.8; CI = 2.2–15.3). ICD9 730.08 encodes for 'acute osteomyelitis, other specified sites'. Manual review of chart records for

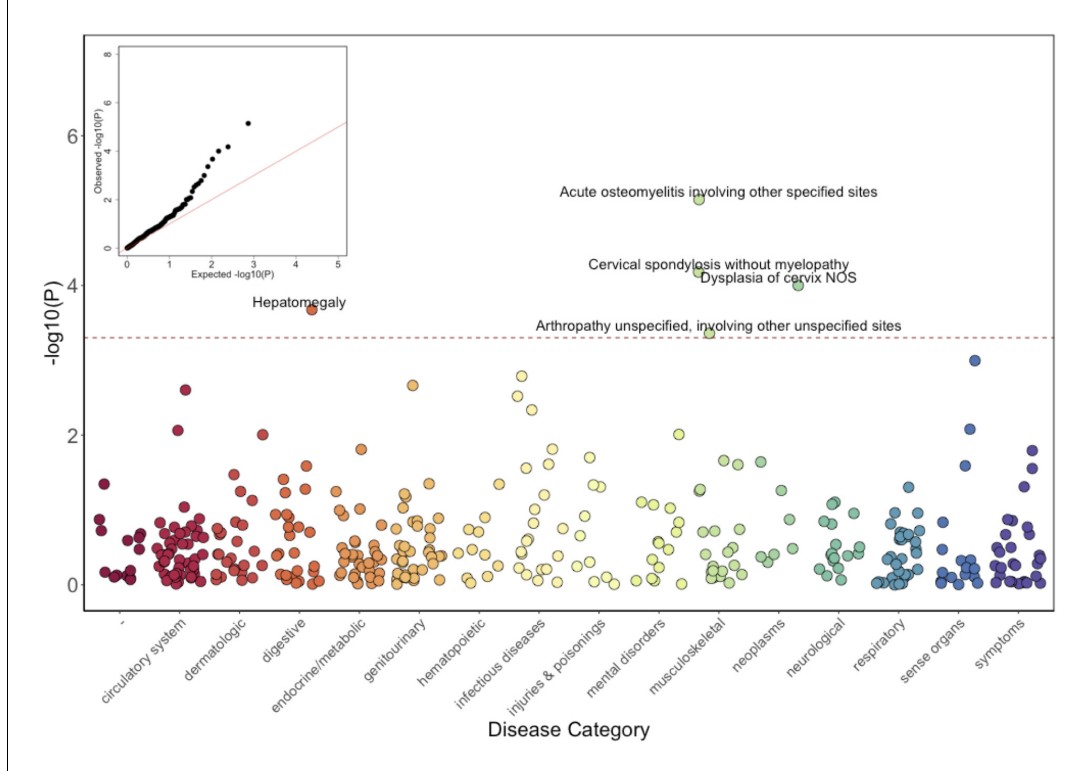

**Figure 3.** Phenome-Wide Association Study (PheWAS) of *COL27A1*.pG697R carriers vs ICD9 billing codes derived from the Electronic Health Records (EHR) under a general linear model (GLM). Five billing codes achieve significance (FDR adjusted p<0.05). Three of the five significant ICD9 codes are in category of musculoskeletal disorders.

DOI: https://doi.org/10.7554/eLife.25060.017

The following figure supplement is available for figure 3:

**Figure supplement 1.** Phenome Wide Association (PheWAS) under three alternative models.

DOI: https://doi.org/10.7554/eLife.25060.018

these patients revealed that this code referred to vertebral osteomyelitis in the three carriers with the ICD9 code. ICD9 721.0 encodes for cervical spondylosis without myelopathy. Cervical spondylosis refers to degenerative changes of the cervical spine, which can eventually progress to encroach on the cervical canal, causing myelopathy (spinal cord injury). A third diagnosis code, 716.98,

**Table 2.** Top five significantly PheWAS associated ICD9 codes in *COL27A1*.pG697R carriers (N = 106) compared to non-carriers (N = 4480).

| Disease category | ICD9 code | Short description | Number of diagnoses among carriers N (%) | Number of diagnoses among non-carriers N (%) | Odds ratio (5% confidence intervals) | P-value |
|---|---|---|---|---|---|---|
| Neoplasms | 622.10 | Dysplasia of cervix, not otherwise specified | 6 (5.7) | 62 (1.4) | 5.4 (2.3–12.6) | $1.0 \times 10^{-4}$ |
| Musculoskeletal | 716.98 | Arthropathy unspecified, involving other unspecified sites | 4 (3.8) | 48 (1.1) | 5.8 (2.2–15.3) | $4.4 \times 10^{-4}$ |
| Musculoskeletal | 721.00 | Cervical spondylosis without myelopathy | 5 (4.5) | 74 (1.6) | 5.4 (2.4–12.3) | $6.6 \times 10^{-5}$ |
| Musculoskeletal | 730.08 | Acute osteomyelitis involving other specified sites | 3 (2.8) | 6 (0.1) | 34.5 (7.4–162) | $7.1 \times 10^{-6}$ |
| Digestive | 789.10 | Hepatomegaly | 3 (2.8) | 17 (0.4) | 11.6 (3.2–42.2) | $2.1 \times 10^{-4}$ |

DOI: https://doi.org/10.7554/eLife.25060.019

encodes for 'arthropathy, unspecified, or involving other specified sites'. Manual review of chart records for these patients revealed that this code referred to knee arthropathy in all four patients. Finally, two other ICD9 codes were significantly associated with the *COL27A1*.pG697R variant; 622.10 ($p_{GLM} < 1 \times 10^{-4}$; OR = 5.4; CI = 2.3–12.6), which encodes for cervical dysplasia, and 789.1 ($p_{GLM} < 2.1 \times 10^{-4}$; OR = 11.6; CI = 3.2–42.2), which encodes for hepatomegaly. Presently, it is unclear whether these two are related to a *COL27A1*.pG697R carrier phenotype, or are spurious associations.

We observed over inflation in the distribution of the PheWAS test statistic, measured by lambda ($\lambda$), for all four score based models ($\lambda_{GLM}$=1.59; $\lambda_{SPATest}$=1.20; $\lambda_{GCTA}$=1.36; $\lambda_{Firth}$=2.09), indicating that no single model fully accounts for the confounding effects of distant relatedness, case:control imbalance and rare variants/ICD9s (*Figure 3—figure supplement 1*). The code linked to vertebral osteomyelitis (730.08) was the top signal in all tests ($p_{SPATest} < 1.4 \times 10^{-4}$; $p_{GCTA} < 7.9 \times 10^{-10}$; $p_{Firth} < 1.5 \times 10^{-9}$), but only remains significant after genomic control adjustment in one of the tests ($p_{GCTA\_adjusted} < 4.6 \times 10^{-5}$). Neither codes linked to cervical spondylosis (721.0; $p_{SPATest} < 3.0 \times 10^{-3}$ (rank=3rd); $p_{GCTA} < 3.3 \times 10^{-3}$ (8th)) or knee arthropathy (716.98; $p_{SPATest} < 0.022$ (21st); $p_{GCTA} < 3.5 \times 10^{-3}$ (9th); $p_{Firth} < 0.001$ (35th))) were significant after genomic control correction. Therefore, while PheWAS analysis provided preliminary support of Steel syndrome-associated clinical features in carriers, best practices for PheWAS models for rare variants/ICD9 codes, and in the presence of population structure, remains an open problem for the genomics community. It is also possible that some relevant clinical features of Steel syndrome might be poorly captured by or absent from medical billing codes.

To evaluate the preliminary evidence from the PheWAS analysis, we performed a second analysis of EHR data that focused on a comprehensive manual chart review to examine for evidence of Steel syndrome characteristics in the *COL27A1*.pG697R carriers in the same manner as performed for homozygotes. We limited the analysis to carriers below the age of 55 (N = 34; mean age 41.8 years) to reduce confounding from age-related related symptoms of spine and joint pain. We also selected 31 age and sex matched Puerto Rican non-carriers for comparison (mean age 40.6 years). Utilizing the same criteria used to characterize Steel syndrome cases, we found no evidence of clinical short stature or hip dislocation in carriers, but did observe a trend of elevated rates of major joint and spine degradation (*Table 1*). In general, 38% (13/34) of carriers showed evidence of spine degeneration varied from severe (multiple level cord compression and neurological symptoms necessitating corrective surgery) to moderate (lower back pain with no neurological symptoms managed with physical therapy and/or pain medication) compared to 13% (4/31) of non-carriers (Fishers exact test p<0.03). Specifically, we found an increased risk of cervical stenosis in 15% (5/34) of carriers compared to 0% (0/31) of controls (p<0.05). Although not reaching statistical significance, we show a trend of 2-fold higher rates of scoliosis (24%; p<0.35), arthritis (38%; p<0.1), and lumbar spine degradation (29%; p<0.25) in carriers compared to non-carriers and previous published reports in similar age groups (*Kebaish et al., 2011*; *Reginster, 2002*). Together these data suggest an appreciable burden of joint and spine degradation in *COL27A1*.pG697R carriers (*Table 1*).

## Worldwide frequency and demographic history of COL27A1.pG69R

Having genetically identified and clinically characterized a previously little-known disease variant, we next investigated which populations were at risk for harboring the allele. We assessed the carrier frequencies of the *COL27A1*.pG697R variant in global panel of 51745 individuals from Africa (N = 376), the Americas (N = 45685), Asia (N = 5311), Europe (N = 209), the Middle East (N = 163) and Oceania (N = 28) genotyped on MEGA in the PAGE Study. This included; 13050 in the Multi-Ethnic Cohort (MEC) (*Kolonel et al., 2000*) Study; 12327 in the Hispanic Community Health Study/Study of Latinos (HCHS/SOL) (*Lavange et al., 2010*); 12852 in the Women's Health Initiative (WHI) Study (*The Women's Health Initiative Study Group, 1998*); 13044 additional Bio*Me* biobank participants (including the 1775 Puerto Ricans on MEGA described above); and a Global Reference Panel from Stanford University including the Human Genome Diversity Panel (*Cann, 2002*) (N = 986, see Materials and methods). Combined, the PAGE and Bio*Me* dataset represented 57316 individuals from 112 global populations (*Supplementary file 4*). The *COL27A1*.pG697R C allele was present in 183 copies (173 heterozygous carriers and 5 homozygous cases). We estimate the carrier rate of *COL27A1*.G697R to be 1:51 in Puerto Rican-born individuals (minor allele frequency (MAF) = 1.1%); 1:9 in individuals born on the island of St. Thomas (MAF = 11%); 1:346 in Hispanic/Latino/as in the

US (MAF = 0.29%) and 1:72 in Bio*Me* Hispanic/Latino/a populations from New York City (MAF = 0.7%); and 1:746 in individuals born in the Dominican Republic (MAF = 0.067%) (*Figure 4A*). We note that only 9 people were assayed from St. Thomas, so the high carrier frequency estimate could be biased by small sample size. Finally, the variant is present in only 4 copies in the 60,706 exomes in the ExAC database (*Exome Aggregation Consortium et al., 2016*), likely due to differences in the populations comprising both datasets.

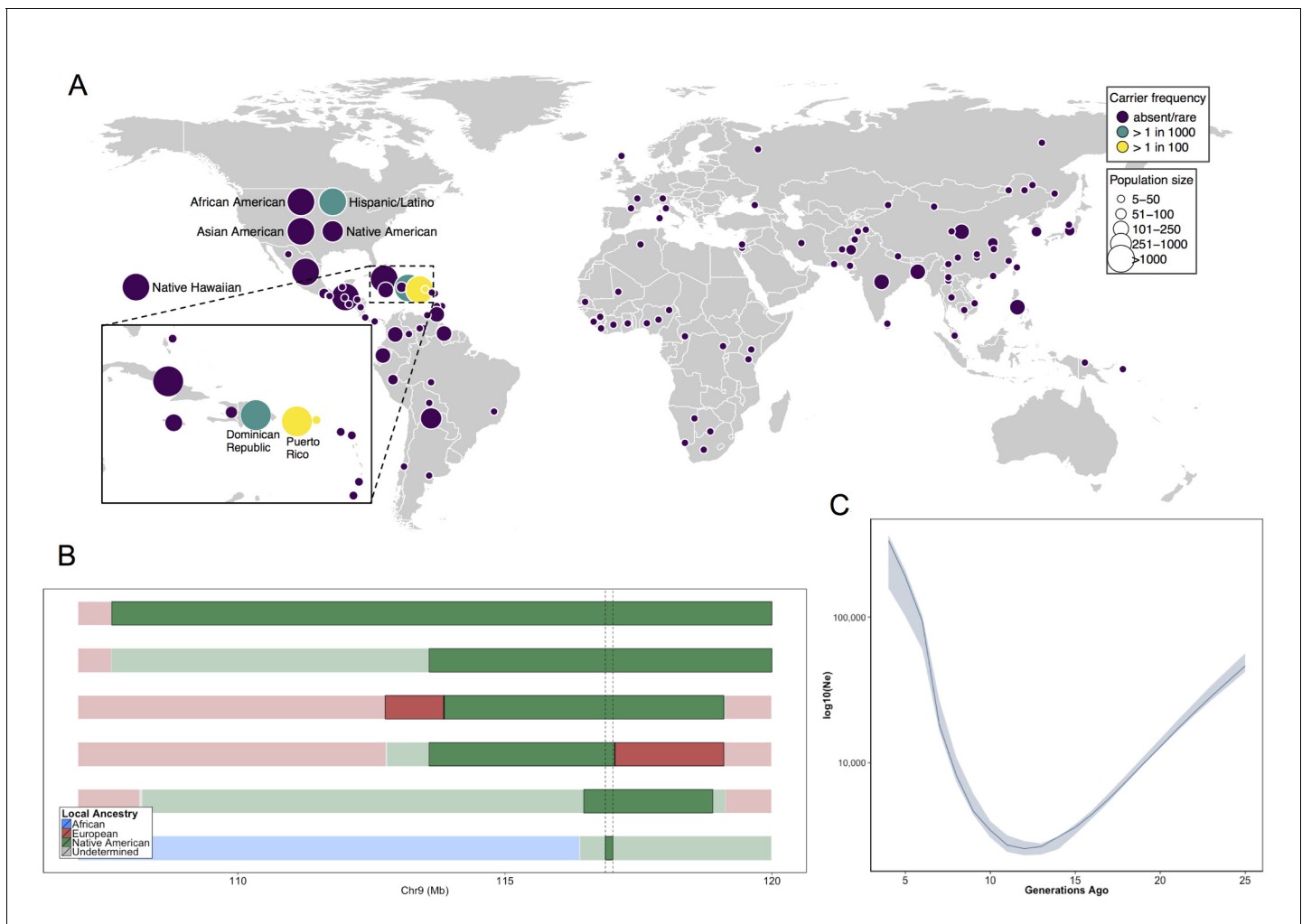

**Figure 4.** Allele frequency distribution and demographic history of *COL27A1*.pG697R. (**A**) Global carrier frequency of *COL27A1*.pG697R in a multi-ethnic database of over 57,000 individuals representing 112 populations. The variant is absent or very rare in most populations (purple), at 1:746 and 1:346 carrier frequency amongst individuals from the Dominican Republic and Hispanic/Latino/as in the United States (green), and at 1:51 and 1:9 carrier frequency amongst individuals from Puerto Rico and St. Thomas (yellow). (**B**) Joint analysis of identity-by-descent and local ancestry haplotypes in three individuals homozygous for the *COL27A1*.pG697R variant. A large 15 cM interval on chromosome 9 is shown with local ancestry inferred as African (blue), European (red) and Native American (green), with shading to indicating the boundaries of the IBD haplotypes. The location of *COL27A1* is indicated by the dashed line (**C**) Effective population size of the Puerto Rican discovery population (N = 2816) over the past 4–25 generations inferred from the tract length distribution of IBD haplotypes suggests that the ancestral population underwent a bottleneck approximately 9–14 generations ago. 95% confidence intervals are represented by blue ribbon.

DOI: https://doi.org/10.7554/eLife.25060.020

The following figure supplements are available for figure 4:

**Figure supplement 1.** Integrated haplotype score (iHS) in Puerto Rican ancestry individuals at the 9q32 Locus reveals no evidence of selection.
DOI: https://doi.org/10.7554/eLife.25060.021

**Figure supplement 2.** Global ancestry proportions for the individuals used as reference samples for RelateAdmix in the Bio*Me* H/L.
DOI: https://doi.org/10.7554/eLife.25060.022

To predict what other populations might be at risk for Steel syndrome, we explored the locus-specific demographic history in carriers of the *COL27A1*.pG697R risk haplotype. First, by visual inspection, we were able to discriminate a single haplotype of 107.5 kb in length that contained 55 SNPs, which uniquely tagged the *COL27A1*.pG697R variant ($R^2$ = 1). This haplotype was present only in individuals born in Puerto Rico (N = 25), the Dominican Republic, (N = 2), Columbia (N = 1), New York City (N = 40) and St. Thomas (N = 1). Genotyping determined that only the haplotype carriers from Puerto Rico, New York City and St. Thomas also carried the *COL27A1*.pG697R variant (N = 56 in total). Second, we inferred continental ancestry along the genomes of the three Puerto Rican homozygotes in the discovery cohort. Local ancestry inference is the task of assigning continental ancestry to genomic segments in an individual with recent ancestors from multiple continents. For the Puerto Rican homozygotes, we estimated local haplotypic similarity with a reference panel of African, European and Native American genomes using RFMix (*Maples et al., 2013*). Examination of local ancestry on the background of the IBD haplotype in all three homozygous individuals revealed all to be homozygous for Native American ancestry, suggesting the *COL27A1*.pG697R arose on a Native American haplotype (*Figure 4B*).

To test whether the disease variant arose via genetic drift or selection, we used the IBDNe software (*Browning and Browning, 2015*) to estimate the historical effective population size ($N_e$) of the Puerto Rican discovery cohort (N = 2816) (*Figure 4C*). The IBDNe software calculates the effective population size of a given population over past generations by modeling the distribution of IBD tract lengths present in the contemporary population. The analysis suggested evidence of a strong bottleneck in Puerto Ricans approximately 9–14 generations ago, with the smallest effective population sized dating approximately 12 generations ago (estimated $N_e$ = 2580, 95% C.I 2320–2910). This is consistent with the timing of European immigration and slave trading on the Island, resulting in admixture and population bottlenecking, followed by demographic growth post-contact (*Moreno-Estrada et al., 2013*; *Gravel et al., 2013*). Finally, to see if there was evidence that the locus had undergone a recent selective sweep we calculated the integrated haplotype score (iHS) (*Voight et al., 2006*; *Szpiech and Hernandez, 2014*) across chromosome 9 in phased genotype data for Bio*Me* Puerto Rican samples, but did not observe evidence of selection at the locus (*Figure 4—figure supplement 1*). Together, this evidence suggests that the *COL27A1*.pG697R variant arose in the ancestral Native American populations that peopled the Caribbean, which underwent a strong bottleneck during the period of colonization, which may help explain the prevalence of this disease in amongst contemporary Puerto Rican populations.

## Discussion

Here we describe a new approach to utilize genomic data in health systems for identifying and characterizing genetic disorders, the cornerstone of which is the ability to identify related individuals in the absence of recorded pedigree or genealogy information. By linking medical records of distantly related patients, identified by shared tracts of genetic homology identical-by-descent (IBD), we discovered a recessive haplotype on 9q32 conferring extreme short stature. Whole genome sequencing revealed that a mutation (Gly697Arg) in the *COL27A1* gene had been previously implicated as the genetic variant underlying Steel syndrome (*Gonzaga-Jauregui et al., 2015*; *Steel et al., 1993*; *Flynn et al., 2010*). Population screening indicated that the disease variant is more common than previously thought in people with Puerto Rican ancestry, and in some other Caribbean populations, and very rare or absent elsewhere in the world. Extensive analysis of clinical records confirms almost all features of the recessive disorder in cases, and reveals potential complications that can occur later in life. An agnostic survey of the medical records of carriers, supplemented by manual chart review, indicates evidence of joint and spine degradation in heterozygotes. Biochemical modeling suggests that *COL27A1*.G697R disrupts a conserved triple helix domain of the alpha-1 collagen in a mechanism similar to dominant forms of other collagen disorders (*Persikov et al., 2004*). Taken together, this study indicates that a single mutation in the *COL27A1* gene underlies a common collagen disorder impacting up to 2% of people of Puerto Rican ancestry.

This is consistent with our finding, supported by previous work (*Moreno-Estrada et al., 2013*), demonstrating a founder effect in Puerto Rican populations. Despite segregating at an estimated carrier rate of 1:51, the *COL27A1*.pG697R variant was first described very recently (*Gonzaga-Jauregui et al., 2015*). This suggests that there may be other highly penetrant disease variants

segregating at appreciable frequencies in Puerto Rican populations (*Anikster et al., 2001*; *Cornier et al., 2008*; *Daniels et al., 2013*; *Al-Zaidy et al., 2015*; *Arnold et al., 2013*; *Lee et al., 2014*), and other understudied founder populations, the discovery of which could lead to new disease variants and biology. Indeed, although *COL27A1* was first implicated as the Steel syndrome disease locus in an extended family from Puerto Rico recently (*Chang et al., 2015*), other variants in *COL27A1* have since been linked to Steel syndrome in Indian (*Kotabagi et al., 2017*) and Emerati (*Gariballa et al., 2017*) families revealing additional clinical features of the disease such as hearing loss. In our own health system, approximately 190,000 patients of Puerto Rican descent are treated annually (*Humes et al., 2011*). We estimate that up to 80 may have the severe homozygous form of the disorder and that the milder heterozygous form could be found in up to 1200 patients. A search of progress notes, discharge summaries, and operative reports of over 4 million patients in the Mount Sinai data warehouse discovered mentions of the text term 'Steel Syndrome' in 42 patient records. However, all of these patients were on dialysis for end stage renal disease, indicating that this mention was a misspelling of vascular Steal Syndrome, which is common in dialysis patients. This suggests that Steel syndrome might be largely undiagnosed. Attempts are currently being made to re-contact Bio*Me* participants with suspected Steel syndrome, and a genetic test is now available at Mount Sinai (website: http://sema4genomics.com/products/test-catalog/).

This study highlights the benefits of incorporating statistical and population genetics approaches in medical genetic research. First, we demonstrated that leveraging distant relationships *via* IBD mapping was better powered for discovery of the *COL27A1* variant compared to a more typical GWAS approach (i.e. genotype, imputation, and SNP association). As sample sizes increase in health systems and biobanks, the odds of a new individual being a direct or distant relative of someone already in the database increases exponentially (*Henn et al., 2012*), enabling the detection of shared haplotypes harboring rarer causal variants and better-powered IBD mapping studies. Second, we inferred that *COL27A1*.pG697R variant arose on a Native American haplotype, and we estimate that the allele may have segregated at a carrier frequency of 25–30% in pre-Columbian Taíno populations and/or been driven to its current frequency by a bottleneck that occurred during the early days of colonization in Puerto Rico. Therefore this study not only helps estimate population attributable risk of *COL27A1*.pG697R in Puerto Rican populations, but also to predict other populations potentially at risk, including other Caribbean and Taíno populations. Targeted population screening of *COL27A1*.pG697R could potentially provide personalized health management, surveillance for associated complications, guidelines for intervention (particularly in newborns [*Flynn et al., 2010*]), and improved reproductive choices.

This work also highlights some of the current challenges in the emerging field of genomic medicine. We demonstrated that evidence from EHRs could be readily extracted and retrospectively used to characterize clinical features of a musculoskeletal genetic disorder. However, features of many other genetic disorders may not be detectable via routine clinical exam, lab tests and radiologics, and may not be amenable to such an approach. Furthermore, statistical methods for population-scale disease variant discovery, which were predominantly developed for cohorts collected for genetic research, may not be optimally calibrated for discovery in patient populations encountered in health systems. Finally, many genetic disorders are very rare, or have more complex genetic underpinnings, which would reduce power for detection using the strategy we have described. However, recent efforts, such as the Precision Medicine Initiative, that focus on the broad adoption of genomics in medicine, combined with international efforts to catalog rare genetic diseases, are primed to increase the rate of incidental genetic diagnosis of disease.

In summary, this work demonstrates the utility of biobanks for exploring full medical phenomes, and highlights the importance of documenting a wider spectrum of genetic disorders, in large and diverse populations of humans. In particular, this method provides a bridge between classical medical genetic methods and those employed in population-level GWAS. Here we note that the *COL27A1* variant is very rare in current large-scale genomic databases used for clinical research. Thus traditional association strategies and ascertainment bias focused on populations of European descent would have failed to identify and characterize this disorder and its public health burden. As ours and other recent studies have demonstrated, EHR-embedded research will be increasingly important for disentangling the pathology of rare genetic disorders, and understanding the continuum of complex and Mendelian disease. As studies grow in size, and healthcare systems learn to leverage the wealth of information captured in the EHR, there is a need to provide relevant medical

information to any patient entering the clinic anywhere in the world. Methods like that described here allow for precision medicine with a truly global outlook.

## Materials and methods

### BioMe biobank program

Study participants were recruited from the Bio*Me* Biobank Program of The Charles Bronfman Institute for Personalized Medicine at Mount Sinai Medical Center from 2007 onward. The Bio*Me* Biobank Program (Institutional Review Board 07–0529) operates under a Mount Sinai Institutional Review Board-approved research protocol. All study participants provided written informed consent. Of the approximately 38000 participants currently enrolled in Bio*Me*, N = 10511 unrelated are genotyped on the Illumina Infinium OmniExpress (OMNI) array. 5102 of these participants self-report as 'Hispanic or Latino/a', 3080 as 'African American or African', 2078 as 'White or Caucasian', 89 as 'Mixed' and 162 as 'Other'. Country of origin information is available for all but N = 5 participants, with N = 6553 reporting being born in the United States and the remaining N = 3953 report being born outside of the US. Parental and grandparental country of origin information is only available for a small subset of individuals genotyped on the OMNI array (N = 43). An additional N = 10471 participants were genotyped on the Illumina Infinium Multi Ethnic Genotyping array (MEGA) v1.0. Of these, approximately 4704 self-reported 'Hispanic or Latino/a' ethnicity, 3143 self-reported as 'African American or African', 22 self-reported as 'White/Caucasian', 708 self-reported as 'Mixed' and 1894 self-reported as 'Other'. Country of birth information was available for all but a small number of participants (N = 228), with 5190 reporting being born in the United States, and the remaining 5053 self-reporting being born elsewhere. Parental and grandparental country of origin information is available for 4323 individuals genotyped on the MEGA array.

### OmniExpress genotyping and QC

Genotyping of 12749 Bio*Me* participants was performed on the Illumina Infinium OmniExpress plus HumanExome array. Calling was performed using the GenomeStudio software. A total of 1093 individuals were removed prior to zCall due to plate failure (N = 672), unambiguous discordance between genetic and EHR recorded sex (N = 693), a call rate of <98% (N = 834), or deviances in levels of heterozygosity (N = 773 in total). This was defined as having either an inbreeding coefficient outside the range −0.1 and 0.3 for common alleles (MAF >1%), or between 0.4 and 0.9 for rare alleles (MAF <1%). Additional quality control of 11656 individuals was performed using PLINK1.7 (*Purcell et al., 2007*) (RRID:SCR_001757). An individual with a call rate of <99% was also excluded (N = 1), along with intentional genetic duplicates (PiHat >0.8, N = 444). Site-level quality control consisted of the removal of SNPs with a call rate of <95% (n = 42217), and the removal of sites that were significantly out of Hardy-Weinberg equilibrium (p<$1\times10^{-5}$; n = 39660) when calculated for self-reported EA, AA and H/L separately. Palindromic sites and those that deviated considerably from the 1KGP allele frequencies (<40% *versus* >60%) were also removed to ensure uniform stranding across datasets. After QC steps, 11212 participants and 866864 SNPs remained for downstream analysis.

### IBS relatedness estimates and estimation of inbreeding Co-efficient

Pairwise IBS-based relationship estimates were derived for Bio*Me* participants (N = 11212) using the RELATEAdmix software (*Moltke and Albrechtsen, 2014*), which accounts for inflation of IBS statistic due to admixture linkage disequillibrium in admixed populations. To include allele frequency information and global ancestry proportions from ancestral populations relevant for each admixed population in the analysis. These were estimated using ADMIXTURE (*Alexander et al., 2009*) (RRID:SCR_001263) H/L samples were merged with the Utah Residents (CEPH) with Northern and Western European Ancestry (CEU; N = 100), Yoruba in Ibadan, Nigeria (YRI; N = 100) and Native American (NA; N = 43; including Nauha, Ayamaran, Mayan and Quechan individuals) and used as input for the ADMIXTURE software, which was run unsupervised at k = 3. ADMIXTURE analysis confirmed that NA reference panel comprised >99% proportion Native American genetic ancestry (*Figure 4—figure supplement 2*). European-American (EA) individuals were merged with the CEU and a panel of self-reported Ashkenazi Jewish individuals genotyped on OMNI from Bio*Me* (AJ; N = 100) and run

unsupervised at k = 2. AA samples were merged with the CEU and YRI reference panels and run unsupervised at k = 2. After intersecting with reference panels 99296 SNPs were used as the input for RELATEAdmix.

## Principal component analysis

Principal Component Analysis (PCA) was performed using the SMARTPCAv10210 software from the EIGENSOFTv5.0.1 (RRID:SCR_004965) (*Price et al., 2006*) in 10511 unrelated Bio*Me* participants. Regions containing the Human Leukocyte Antigen (chr6: 27000000–35000000 (NCBI37/hg19)), Lactase gene (chr2:135000000–137000000 (NCBI37/hg19)) and a common inversion (chr8:6000000–16000000 (NCBI37/hg19)), all of which are regions known to confound PCA analysis were removed from the genotype data prior to analysis. Data were merged with a reference panel of 2504 individuals from Phase 3 of the 1KGP (RRID:SCR_006828) (*The 1000 Genomes Project Consortium, 2015*) that was constructed by extracting OmniExpress sites from whole-genome sequence data. Following this, a further two other relevant reference panels were added: the NA (N = 43), and AJ (N = 100) panels described above. A total of 174468 SNPs remained after intersecting the data with these reference panels.

## Identity-by-descent tract inference and clustering

Phased genotype data were filtered to MAF >0.01 and converted to PLINK format using the FCGENE software (*Roshyara and Scholz, 2014*) (we avoided using PLINK software for the conversion process in order to retain the phase information). Recombination maps from HapMap II (Build GRCh37/hg19) were intersected with the genotyped sites (n = 490510 SNPs). GERMLINE (RRID: SCR_001720) (*Gusev et al., 2009*) was used to infer tracts of identity by descent >3 cM across all pairs of Bio*Me* individuals (N = 11212) using the following flags: '-min_m 3 -err_hom 0 -err_het 2 -bits 25 –haploid'. IBD haplotypes that fell within or overlapped with centromeres, telomeres and regions of low complexity were removed from the GERMLINE output using an in-house Ruby script. Additional quality control measures consisted of the exclusion of regions of the genome where the depth of IBD-sharing (that is, the number of pairwise IBD-haplotypes that contain a given locus of the genome) exceeded 4 standard deviations from the genome-wide mean (*Figure 2—figure supplement 6*).

IBD clustering to identify 'cliques' of three or more IBD haplotypes shared between multiple individuals was then performed using the efficient connect-component-based clustering version of the Dash Associated Shared Haplotypes algorithm (DASH) (*Gusev et al., 2011*), using the default parameters. As a further quality control measure IBD-sharing 'cliques' inferred by DASH that exhibited excessive sharing (which we defined as clique membership that exceeded 4 s.d. above the genome-wide mean) were removed (*Figure 2—figure supplement 7*). Data was outputted from DASH in PLINK tped format, and alleles were encoded as; homozygote member in a clique as '2', heterozygote member as '1' and everyone else not a member in the clique encoded as '0'.

## Population-level IBD sharing

We calculated the length of any pairwise IBD tract (or sum of the lengths if a pair of individuals shared more than one tract IBD) for each IBD sharing pair within each population to obtain an estimate of the mean and variance of pairwise sharing per population. To compare the tract length distribution between populations (of size *N*), we first binned pairwise IBD tracts by length bin in 0.01 cM increments. We then summed the number of pairwise IBD tracts falling into each length bin (*x*), and divided this number by the number of possible pairwise IBD sharing for each population: N*(N-1)/2.

## Height measurement and transformation

A self-reported measurement of height in feet and inches was recorded for each participant at enrollment into the Bio*Me* program. Raw height data were stratified on the basis of sex for all individuals who were inferred to be of Puerto Rican ancestry (N = 2816). Height data was then log transformed and converted to age-adjusted Z-scores. Participants were excluded on the basis of age reported at the point of enrollment, with a minimum cut-off of 18 years old for females (N = 0) and

22 for males (N = 0), and a maximum cut off of 79 years old for both sexes (N = 194) leaving a total of n = 2622 PR.

## Association of IBD-cliques with height under a recessive model

Association of IBD clique membership with height as a continuous trait was performed under a recessive model using PLINKv1.9 (*Chang et al., 2015*) using the '*–linear recessive*' flag. Age and sex adjusted Z-scores for height were used as the outcome variable. IBD clique membership was used as the primary predictor variable and the first five PCA eigenvectors were used as covariates. The model was run across a total of 2622 PR ancestry individuals and a total of 480 IBD-cliques where at least 3 individuals were homozygous for the IBD haplotype.

## Genome wide association of imputed data under a recessive model

Genotype data for all of the Bio*Me* individuals ascertained on the Illumina OMNI Express array (N = 11212) were phased together using SHAPEIT2 (*Delaneau et al., 2011*; *O'Connell et al., 2014*). Imputation was subsequently performed in 5 MB chunks using IMPUTE2 (RRID:SCR_013055) (*Howie et al., 2009*)*via* the flags '*-Ne 20000 -buffer 250 -filt_rules_l 'ALL<0.0002' 'ALL>0.9998''* with a reference panel derived from phase 3 data from the 1KGP. A total of 46538253 SNPs were imputed from 828109 directly genotyped SNPs.

We ran a recessive GWAS on the same 2622 inferred Puerto Rican ancestry individuals used in our recessive IBD-mapping model. The association was run over hard-called data using the PLINKv1.9 software using the '*–linear recessive*' flag. Age and sex adjusted Z-scores for height were used as the phenotypic outcome and the first five PC eigenvectors were used as covariates. Analysis was restricted to SNPs with >= 2 observations of individuals homozygous for the minor allele (as the only 2 of the 3 homozygotes had been imputed correctly), and SNPs with an INFO score of >= 0.3 (n = 10007795 SNPs in total).

## Whole genome sequencing

Genomic libraries were prepared from DNA obtained for the four IBD homozygous individuals. DNA was sheared to 300 bp on a Covaris E220, libraries were made using the NEBNext Ultra DNA Library Prep kit for Illumina. The libraries were submitted for Whole Genome Sequencing (WGS) at the Mount Sinai Genomic Core using the Illumina HiSeq 2500 system, performed by the Genomics Core Facility of the Icahn Institute for Genomics and Multiscale Biology, Icahn School of Medicine at Mount Sinai. Reads were aligned to the NCBI37/hg19 reference genome and variants were called using the sequence analysis pipeline by *Linderman et al. (2014)* Variant calls and coverage at every site at the genomic interval spanned by the candidate IBD haplotype (chr9:112000000–118000000 bp (NCBI37/hg19)) were obtained using the '*-out_mode EMIT_ALL_SITES*' flag in GATKv3.2–72 (RRID:SCR_001876). For summary statistics of whole genome sequencing (WGS) see *Supplement file 1*.

## In silico analysis and validation of COL27A1.pG967R

WGS variant calls were annotated with allele frequency information and in silico prediction scores for SIFT, PhyloP, GERP generated using snpEffv3.0 (RRID:SCR_005191) as part of the sequence analysis pipeline published by *Linderman et al. (2014)*. We identified all genomic variants that were present in at least 6 copies across the four IBD-homozygotes and that lay within the shared boundary of the IBD haplotype. Using this criteria, only one rare, coding variant was found to be shared between all four homozygotes, namely a point mutation in the gene *COL27A1* (g.9:116958257 .C>G, NM_032888.1, p.G697R, rs140950220) which was present in 7 copies (with 3 individuals being homozygous, and the fourth being heterozygous). The rs140950220 G/C allele status was validated by Sanger sequencing of exon 7 in the *COL27A1* gene in all four individuals. We also validated *COL27A1*.pG697R status in individuals carrying the significant IBD-clique at 9q32 using the Fluidigm SNPType assay adhering to the standard protocol. All individuals carrying at least one copy of the top IBD-haplotype (N = 59) were genotyped for the rs140950220 variant in addition to a panel of age and sex matched Puerto Rican ancestry controls (N = 59).

## Genotyping COL27A1.pG697R in a Multi-Ethnic population of PAGE

We estimated the frequency of the *COL27A1.pG697R* (dbSNP = rs140950220) variant in the Population Architecture using Genomics and Epidemiology (PAGE) study. The PAGE study comprises a diverse global reference panel from five studies. African-American and Hispanic/Latina women from the Women's Health Initiative (WHI), a multi-center cohort study investigating post-menopausal women's health in the US and recruited women at 40 centers across the US. Self-identified Hispanic/Latino/as from four sites in San Diego, CA, Chicago, IL, Bronx, NY, and Miami, FL as part of the Hispanic Community Health Study/Study of Latinos (HCHS/SOL). African American, Japanese American, and Native Hawaiian participants from the Multiethnic Cohort (MEC) prospective cohort study recruiting men and women from Hawaii and California. The Global Reference Panel (GRP) created by Stanford University contributed samples including; a population sample of Andean individuals primarily of Quechuan/Aymaran ancestry from Puno, Peru; a population sample of Easter Island (Rapa Nui), Chile; individuals of indigenous origin from Oaxaca, Mexico, Honduras, Colombia, the Nama and Khomani KhoeSan populations of the Northern Cape, South Africa; the Human Genome Diversity Panel in collaboration with the Centre Etude Polymorphism Humain (CEPH) in Paris; and the Maasai in Kinyawa, Kenya (MKK) dataset from the International Hapmap Project hosted at Coriell. Finally, the Bio*Me* biobank in the Mount Sinai health system, New York City, contributed African-American, Hispanic/Latino/a, and participants who reported as mixed or other ancestry to the PAGE study,~50% of whom were born outside New York City and for whom country-of-birth information was available. In all, participants in the PAGE Study represent a global reference panel of 112 populations ranging from 4 to 17773 individuals in size (*Supplement file 4*). Samples in the PAGE study were genotyped on the Illumina Multi-Ethnic Genotyping Array (MEGA), which included direct genotyping of the rs140950220 variant. A total of 53338 PAGE and GRP samples were genotyped on the MEGA array at the Johns Hopkins Center for Inherited Disease Research (CIDR), with 52878 samples successfully passing CIDR's QC process. Genotyping data that passed initial quality control at CIDR were released to the Quality Assurance/Quality Control (QA/QC) analysis team at the University of Washington Genetics Coordinating Center (UWGCC). The UWGCC further cleaned the data according to previously described methods (*Laurie et al., 2010*) and returned genotypes for 51520 subjects. A total of 1705969 SNPs were genotyped on the MEGA. The *COL27A1*.pG697R variant passed the following filters; (1) CIDR technical filters, (2) SNPs with missing call rate >= 2%, (3) SNPs with more than 6 discordant calls in 988 study duplicates, (4) SNPs with greater than 1 Mendelian errors in 282 trios and 1439 duos, (5) SNPs with a Hardy-Weinberg $p<10^{-10}$, (6) positional duplicates.

## Structural modeling of the COL27A1.PG697R missense variant

We downloaded X-ray crystal coordinates (1CAG from *Bella et al., 1994*.; www.pdb.org) on January 21, 2017. Visualization and modeling of the missense variant were performed in PyMol (www.pymol.org; RRID:SCR_000305).

## Phenome-Wide association study

To test for clinical symptoms of Steel syndrome in *COL27A1*.pG967R carriers, we performed a Phenome-Wide Association Study (PheWas) with EHR-derived ICD9 billing codes as the phenotypic outcome. In the association model, for each individual ICD9 codes were encoded as '1' if the ICD9 was present in their EHR, and '0' if the ICD9 code was absent. Carrier status for *COL27A1*.pG697R was used as the primary predictor variable, with heterozygous individuals encoded as '1', non-carriers encoded a '0' and homozygotes excluded from the analysis. We restricted the analysis to carriers of *COL27A1*.pG697R (n = 106) and non-carriers (n = 4480) who either reported being born in Puerto Rico or who were US-born, self-identified as H/L and overlapped with Puerto Rican born individuals in principal component analysis. Age, sex and the first 5 principal components were included as covariates in our model. The regression was performed using four methods; Generalized Linear Models (GLM) using the glm() function in Rv3.2.1; a score test based on the saddlepoint approximation (SPATest) using the SPAtest() function in Rv3.2.1; a score test using a base adjustment for rare variants (Firth test) using the logistf() function in Rv3.2.1; and a linear mixed model using the GCTAv1.24.2 software with a genetic relationship matrix constructed from 281666 SNPs shared between the OMNI and MEGA arrays (MAF >= 1%). To adjust for multiple tests, raw p-values were

adjusted for false discovery rate using the p.adjust() function in R, and only those below an FDR adjusted p-value of 0.05 were reported as significant.

## Clinical review of patient records

Information from inpatient, outpatient, emergency and private practice settings housed in the Mount Sinai health system since 2004 was reviewed by two clinical experts independently. This data includes laboratory reports, radiological data, pathology results, operative and inpatient/outpatient progress notes, discharge summaries, pharmacy, and nurses reports. The clinical experts examined for clinical features similar to those reported for Steel syndrome cases in *Flynn et al. (2010)*, including developmental dysplasia of the hip (or congenital hip dysplasia), carpal coalition, scoliosis, and joint and spine anomalies. Both clinical experts reviews patient records independently and compared notes to resolve discrepancies. They reviewed the records of the 34 youngest *COL27A1*.pG697R carriers (mean age 42), and compared their findings to 31 randomly selected age and sex matched Puerto Rican non-carriers, and also to published reports of population prevalences of key clinical features for similar age groups where available.

## Local ancestry estimation

Due to the process of recombination, individuals from populations that have undergone recent admixture can exhibit a mosaic of genetic ancestry along their genome. Their genetic ancestry at a given genomic segment (referred to as local ancestry), can be inferred from genotype data with the use of non-admixed reference panels of known continental ancestry. We calculated local ancestry in the three homozygous Puerto Rican individuals genotyped on OMNI by first extracting the intersecting sites of the Affymetrix 6.0 array (n = 593729 SNPs in total) and merging them with 3 ancestral reference panels. These reference panels consisted of the CEU and YRI samples from the 1KGP in addition to the Native American reference panel described previously that were used as a proxy for European, African and Native American ancestral source populations, respectively. RFMix (*Maples et al., 2013*) was used to infer local ancestry.

## Calculation of historical effective population size in Puerto Ricans

To investigate evidence of a founder effect in Puerto Ricans we ran the IBDNe software (*Browning and Browning, 2015*) in 2816 Puerto Ricans from the discovery effort using the cleaned set of pairwise IBD-haplotypes inferred using GERMLINE. IBDNe was run using the default parameters, including an assumed generation time of 25 years.

## Data availability

All scripts used to generate main and supplementary figures for this manuscript are available for download on GitHub (https://github.com/gillian-belbin/ibdmapping_ehr_figscripts [*Belbin, 2017*]; a copy is archived at https://github.com/elifesciences-publications/ibdmapping_ehr_figscripts). The data frames of analysis results that were used to generate the main and supplementary figures are located here: http://research.mssm.edu/kennylab/data_source_file.tar.gz

## Web resources

BioMe OmniExpress data: https://www.ncbi.nlm.nih.gov/projects/gap/cgi-bin/study.cgi?study_id=phs000888.v1.p1

BioMe MEGA data: https://www.ncbi.nlm.nih.gov/projects/gap/cgi-bin/study.cgi?study_id=phs000925

PAGE MEGA data: https://www.ncbi.nlm.nih.gov/projects/gap/cgi-bin/study.cgi?study_id=phs000356

*Location of Native American panels:* ftp://ftp.1000genomes.ebi.ac.uk/vol1/ftp/technical/working/20130711_native_american_admix_train/

Software used in the analysis:

SMARTPCA: https://github.com/DReichLab/EIG

ADMIXTURE: https://www.genetics.ucla.edu/software/admixture/download.html

RelateAdmix: http://www.popgen.dk/software/index.php/RelateAdmix#Download

RFMix: https://sites.google.com/site/rfmixlocalancestryinference/

GERMLINE: http://www.cs.columbia.edu/~gusev/germline/
DASH: http://www1.cs.columbia.edu/~gusev/dash/
PyMol: www.pymol.org
IBDNe: http://faculty.washington.edu/browning/ibdne.html

## Acknowledgements

We would like to thank Noah Zaitlen, Alexander Gusev, George Diaz, Rounak Dey, and Alicia Martin for their helpful suggestions in preparing this manuscript. This work was supported by funds from several grants; The Population Architecture Using Genomics and Epidemiology (PAGE) program is funded by the National Human Genome Research Institute, with co-funding from the National Institute on Minority Health and Health Disparities, supported by U01HG007416, U01HG007417, U01HG007397, U01HG007376, and U01HG007419. The PAGE consortium thanks the staff and participants of all PAGE studies for their important contributions. The complete list of PAGE members can be found at http://www.pagestudy.org. Genotyping services were provided by the Center for Inherited Disease Research (CIDR). CIDR is fully funded through a federal contract from the National Institutes of Health to The Johns Hopkins University, contract number HHSN268201200008I. Genotype data quality control and quality assurance services were provided by the Genetic Analysis Center in the Biostatistics Department of the University of Washington, through support provided by the CIDR contract. High performance computing was supported in part through the computational resources and staff expertise provided by Scientific Computing at the Icahn School of Medicine at Mount Sinai and through the Office of Research Infrastructure under award number S10OD018522. The content is solely the responsibility of the authors and does not necessarily represent the official views of the National Institutes of Health.

## Additional information

### Competing interests

Ruth JF Loos: Reviewing editor, *eLife*. The other authors declare that no competing interests exist.

### Funding

| Funder | Grant reference number | Author |
|---|---|---|
| National Human Genome Research Institute | U01HG009080 | Gillian Morven Belbin<br>Sumita Kohli<br>Eimear E Kenny |
| National Human Genome Research Institute | U01HG109391 | Gillian Morven Belbin<br>Sumita Kohli<br>Eimear E Kenny |
| National Human Genome Research Institute | U01HG007416 | Gillian Morven Belbin<br>Claudia Schurmann<br>Ruth Loos<br>Eimear E Kenny |
| National Human Genome Research Institute | U01HG007417 | Anne E Justice<br>Kristin L Young<br>Misa Graff<br>Kari E North |
| National Human Genome Research Institute | U01HG007397 | Christopher R Gignoux<br>Genevieve L Wojcik |
| National Human Genome Research Institute | U01HG007376 | Ulrike Peters |

| National Institute on Minority Health and Health Disparities | HHSN268201200008 | Gillian Morven Belbin<br>Elena P Sorokin<br>Christopher R Gignoux<br>Genevieve L Wojcik<br>Misa Graff<br>Kari E North<br>Ulrike Peters<br>Ruth Loos<br>Eimear E Kenny |
|---|---|---|
| National Human Genome Research Institute | HHSN268201200008 | Gillian Morven Belbin<br>Elena P Sorokin<br>Sumita Kohli<br>Christopher R Gignoux<br>Genevieve L Wojcik<br>Misa Graff<br>Kari E North<br>Ulrike Peters<br>Ruth Loos<br>Eimear E Kenny |

The funders had no role in study design, data collection and interpretation, or the decision to submit the work for publication.

## Author contributions

Gillian Morven Belbin, Conceptualization, Data curation, Software, Formal analysis, Supervision, Visualization, Methodology, Writing—original draft, Writing—review and editing; Jacqueline Odgis, Sumita Kohli, Formal analysis, Validation, Writing—review and editing; Elena P Sorokin, Formal analysis, Visualization, Writing—review and editing; Muh-Ching Yee, Validation, Methodology, Writing—review and editing; Benjamin S Glicksberg, Formal analysis, Writing—review and editing; Christopher R Gignoux, Genevieve L Wojcik, Investigation, Writing—review and editing; Tielman Van Vleck, Software, Validation, Investigation; Janina M Jeff, Data curation, Methodology; Michael Linderman, Resources, Software; Claudia Schurmann, Douglas Ruderfer, Resources, Data curation, Methodology; Xiaoqiang Cai, Ruth Kornreich, Validation, Methodology; Amanda Merkelson, Data curation, Methodology, Project administration; Anne E Justice, Kristin L Young, Misa Graff, Data curation, Writing—review and editing; Kari E North, Ulrike Peters, Regina James, Lucia Hindorff, Project administration, Writing—review and editing; Lisa Edelmann, Validation, Methodology, Project administration; Omri Gottesman, Conceptualization, Resources, Data curation; Eli EA Stahl, Resources, Formal analysis; Judy H Cho, Resources, Project administration, Writing—review and editing; Ruth JF Loos, Resources, Data curation, Project administration, Writing—review and editing; Erwin P Bottinger, Conceptualization, Resources, Project administration; Girish N Nadkarni, Formal analysis, Supervision, Writing—review and editing; Noura S Abul-Husn, Formal analysis, Validation, Writing—original draft; Eimear E Kenny, Conceptualization, Formal analysis, Supervision, Writing—original draft, Writing—review and editing

## Author ORCIDs

Gillian Morven Belbin http://orcid.org/0000-0001-9036-864X
Eimear E Kenny http://orcid.org/0000-0001-9198-759X

## Ethics

Human subjects: Study participants were recruited from the BioMe Biobank Program of The Charles Bronfman Institute for Personalized Medicine at Mount Sinai Medical Center from 2007 onward. The BioMe Biobank Program (Institutional Review Board 07-0529) operates under a Mount Sinai Institutional Review Board-approved research protocol. All study participants provided written informed consent.

## Decision letter and Author response

Decision letter https://doi.org/10.7554/eLife.25060.032
Author response https://doi.org/10.7554/eLife.25060.033

## Additional files

### Supplementary files

• Supplementary file 1. Summary statistics for 4-18X whole genome sequencing in 4 homozygotes
DOI: https://doi.org/10.7554/eLife.25060.023

• Supplementary file 2. Summary of variants on chromosome 9 at the identity-by-descent mapping interval 112–120 cM.
DOI: https://doi.org/10.7554/eLife.25060.024

• Supplementary file 3. Concordance of true *COL27A1*.pG697R genotype status with IBD-clique membership and imputed variant.
DOI: https://doi.org/10.7554/eLife.25060.025

• Supplementary file 4. Summary of COL27A1.p697R in a Multi-ethnic Population in over 50,000 Individuals from 112 global populations. Populations indicated in italics are from the Human Genomes Diversity Panel, all others were genotyped as part of the Population Architecture using Genomics and Epidemiology Study.
DOI: https://doi.org/10.7554/eLife.25060.026

• Transparent reporting form
DOI: https://doi.org/10.7554/eLife.25060.027

### Major datasets

The following previously published datasets were used:

| Author(s) | Year | Dataset title | Dataset URL | Database, license, and accessibility information |
|---|---|---|---|---|
| Tayo BO, Teil M, Tong L, Qin H, Khitrov G, Zhang W, Song Q, Gottesman O, Zhu X, Pereira AC, Cooper RS, Bottinger EP | 2011 | Charles R. Bronfman Institute for Personalized Medicine (IPM) BioBank Genome Wide Association Study of Cardiovascular, Renal and Metabolic Phenotypes | https://www.ncbi.nlm.nih.gov/projects/gap/cgi-bin/study.cgi?study_id=phs000388.v1.p1 | Publicly available at NCBI dbGaP (accession no: phs000388.v1.p1) |
| PAGE Study | 2016 | Population Architecture using Genomics and Epidemiology (PAGE) | https://www.ncbi.nlm.nih.gov/projects/gap/cgi-bin/study.cgi?study_id=phs000356 | Publicly available at NCBI dbGaP (accession no: phs000356.v1.p1) |

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
