## [Decision Letter]

Thank you for submitting your article "Genetic Identification of a Common Collagen Disease in Puerto Ricans via Identity-by-Descent Mapping in a Health System" for consideration by *eLife*. Your article has been reviewed by two peer reviewers, and the evaluation has been overseen by a Reviewing Editor and Mark McCarthy as the Senior Editor. The reviewers have opted to remain anonymous.

The reviewers have discussed the reviews with one another and the Reviewing Editor has drafted this decision letter to help you prepare a revised submission.

Summary:

The *eLife* editorial team finds your submitted research paper of considerable interest: it provides an excellent example how a large and clinically well characterised cohort with special population history can provide improved power to find novel rare mutations and haplotypes linked to Mendelian syndromes. Furthermore, the presented analytical strategy improves our understanding of already reported pathogenic mutations by efficiently increasing the number of variant carriers and finessing the resulting clinical picture.

The various contributors to the *eLife* Editorial team (reviewers, Reviewing Editor, Senior Editor) have raised some questions about aspects of the statistical and methodological approaches, and identified some weaknesses in the methods description, reported sample sizes and in referencing to tables and figures. Furthermore, we felt that some aspects of the work, like IBD detection, need to be explained more effectively to non-expert *eLife* readers.

Below, we provide a condensed list of Essential Revision actions that we consider need to be addressed in any revised submission.

Essential revisions:

1) Aspects around IBD – Please provide a more thorough description on the IBD methodology used. We'd recommend starting with a more general description of IBD and its role in causal mutation detection – currently the text is drafted in a manner that almost assumes the reader is very familiar with IBD analysis. Please explain specific terms, like "local ancestry".

Second, please justify the use of 3cM in IBD mapping. Third, provide more details in "Materials and methods" on how IBD cliques were identified and QCd.

2) Aspects around Statistical Methods – Currently, the manuscript lacks clarity on exactly which statistical models were applied to the data.

First, be specific about the regression models referred to in paragraph two of subsection “Detecting a Locus Shared Identical-by-Descent Underlying Extreme Short Stature in 172 Puerto Ricans” – have you modelled the zscore as a function of the number of copies (0, 1, or 2) of the haplotype in a specific clique?

Second, the initial analysis of cliques relied on permutation to evaluate significance. Then when restricting analysis to Puerto Rican ancestry subjects (paragraph three of subsection “Detecting a Locus Shared Identical-by-Descent Underlying Extreme Short Stature in 172 Puerto Ricans”) and 480 cliques w/ at least 3 people, are you using significance evaluated from asymptotics? Why the change in methodology? Bonferroni thresholds are mentioned, correcting for how many tests?

Third, the analysis of ICD9 codes vs. genotype at COL27A1.pG697R – we assume you did 416 different logistic regressions, coding each subject as yes/no for a particular ICD9 diagnosis – this could be made more clear. Was any correction done here for 416 tests? While the significance values are impressive, we concerned about inflation due to case-control imbalance.

Fourth, please repeat ICD9 codes vs. genotype at COL27A1.pG697R association testing using Firth test. It appears that many of these dx categories are quite rare, as is the variant. Logistic regression with case-control imbalance results in serious inflation of test statistics, particularly in the case of rare variants (see Ma et al., 2013 Genetic Epidemiology).

3) Power of IBD mapping – First, currently no power calculations are presented to substantiate the claim that the IBD mapping approach outperforms GWAS in terms of power. Please present these calculations as the p-value yielded through 10,000 permutations is only p=0.0018, which is substantially less than a GWAS significance threshold. Second, please clarify if the COL27A1.pG697R variant is imputable with 1000 Genomes reference. In the Materials and methods section, you provide detailed description on imputation. Furthermore in paragraph two of subsection “Fine-mapping Short Stature Locus Reveals Putative Link to Mendelian Syndrome” it states that there was 96.7% concordance between variant genotype and the associated haplotype – so wouldn't the GWAS have identified the association using the variant?

4) Genetic ancestry and COL27A1.pG697R variant – Currently the text is lacking detailed description of the pathogenic variant. Furthermore, more detail is essential to understand how partitioning into ancestry groups was performed.

First, in the functional analysis section (subsection “Functional Investigation of COL27A1.pG697R”). Is the variant predicted damaging by popular annotation tools?- polyphen, sift, cadd, gerp, mutation taster, phyloP etc.

Second, please provide a confidence interval for the dating of the variant, (around 36 generations).

Third, please provide exact criteria how ancestry groups were defined from PCA vectors. Such details are currently missing from the Materials and methods. Furthermore, please provide clear evidence that suggest the populations that are described in subsection “Local Ancestry Estimation” present Native American are not admixed for European ancestry, or at least not significantly so.

---

## [Author Response]

1) Aspects around IBD – Please provide a more thorough description on the IBD methodology used. We'd recommend starting with a more general description of IBD and its role in causal mutation detection – currently the text is drafted in a manner that almost assumes the reader is very familiar with IBD analysis. Please explain specific terms, like "local ancestry".Second, please justify the use of 3cM in IBD mapping. Third, provide more details in "Materials and methods" on how IBD cliques were identified and QCd.

We have updated the text in several ways to address these points. We added a description of IBD mapping to the Introduction, Results, including references to previously published IBD mapping papers, Discussion, and Materials and methods. We have also added a supplemental figure schematic outlining the IBD mapping pipeline (Figure 1—figure supplement 2), referenced in the Results section. We have added explanation and background for the term local ancestry in several places throughout the text including Results and Materials and methods sections. We have added an explanation to justify the 3cM minimum used in IBD in the Results, including references that address this issue. Finally, we provide more details on how IBD cliques were identified and QC’d in Materials and methods, and Figure 2—figure supplement 6 and Figure 2—figure supplement 7.

2) Aspects around Statistical Methods – Currently, the manuscript lacks clarity on exactly which statistical models were applied to the data.First, be specific about the regression models referred to in paragraph two of subsection “Detecting a Locus Shared Identical-by-Descent Underlying Extreme Short Stature in 172 Puerto Ricans” – have you modelled the zscore as a function of the number of copies (0, 1, or 2) of the haplotype in a specific clique?Second, the initial analysis of cliques relied on permutation to evaluate significance. Then when restricting analysis to Puerto Rican ancestry subjects (paragraph three of subsection “Detecting a Locus Shared Identical-by-Descent Underlying Extreme Short Stature in 172 Puerto Ricans”) and 480 cliques w/ at least 3 people, are you using significance evaluated from asymptotics? Why the change in methodology? Bonferroni thresholds are mentioned, correcting for how many tests?Third, the analysis of ICD9 codes vs. genotype at COL27A1.pG697R – we assume you did 416 different logistic regressions, coding each subject as yes/no for a particular ICD9 diagnosis – this could be made more clear. Was any correction done here for 416 tests? While the significance values are impressive, we concerned about inflation due to case-control imbalance.Fourth, please repeat ICD9 codes vs. genotype at COL27A1.pG697R association testing using Firth test. It appears that many of these dx categories are quite rare, as is the variant. Logistic regression with case-control imbalance results in serious inflation of test statistics, particularly in the case of rare variants (see Ma et al., 2013 Genetic Epidemiology).

We have revised the text substantially to update and clarify the details of the analysis, removed discussion of the additive IBD analysis as it does not add substantially to the our findings, and reordered the presentation of the analysis steps for better comprehension. To address the first point, we have explicitly described how we modeled the z scores in the Materials and method section. For the second point, we defined the Bonferonni adjustment in the Results and report Bonferonni adjusted genome-wide significance. To address the third point, we have clarified the description of the PheWAS analysis in the Results and Material and methods sections.

Regarding the fourth and fifth point, we agree with the reviewers general concern, however best practices for PheWAS models that jointly account for case:control imbalance, rare variants/ICD9 codes, and elevated distant relatedness remain an open problem for the genomics community. At the reviewers suggestion, and to examine this question, in addition to running an Generalized Linear Model (GLM), we ran three other score based tests; the Firth test that incorporates a bias-reduction for small numbers of observations (Ma et al., Genetic Epi 2013); a linear mixed model (GCTA) to account for distant relatedness (Yang et al., 2011); and a test that uses saddlepoint approximation (SPATest) to account for case:control imbalance (Dey et al., 2017). GLM and Firth test are each well vetted methods for case:control association, and GLMs are extensively used in PheWAS analysis. We also ran GCTA in binary trait mode and SPATest, which the author have kindly made available for download pre-publication. We added age, sex and the first five PC eigenvectors as covariates to each model. As anticipated, we observed over inflation in the distribution of the PheWAS test statistic, measured by λ (λ), for all four score based models (λGLM=1.59; λSPATest=1.20; λGCTA=1.36; λFirth=2.09), indicating that none of these models is perfectly calibrated. The top signal we originally reported, the ICD9 code linked to vertebral osteomyelitis (730.08), was the top signal in all four models. The other two signals, cervical spondylosis (721.0) and knee arthropathy (716.98) were each highly ranked, but only significantly in two of the four models. Therefore, to investigate whether there was support for the PheWAS findings, we extended the manual analysis of patient medical records, representing hundreds of hours of manual clinical chart review by two independent clinical experts. Information from inpatient, outpatient, emergency and private practice settings housed in the Mount Sinai health system since 2004 was reviewed. This data includes laboratory reports, radiological data, pathology results, operative notes and discharge summaries, pharmacy, inpatient results reviews, and nurses reports. The clinical experts examined for clinical features similar to those reported for Steel syndrome cases in Flynn et al., 2010, including developmental dysplasia of the hip (or congenital hip dysplasia), carpal coalition, scoliosis, and joint and spine anomalies. Both clinical experts reviewed patient records independently and compared notes to resolve discrepancies. They reviewed the records of the 34 youngest COL27A1.pG697R carriers (mean age 42), and compared their findings to 31 randomly selected age and sex matched Puerto Rican non-carriers, and also to published reports of population prevalences of key clinical features for similar age groups where available. In general they found increased incidence of strong to moderate spine and joint degradation in carriers compare to noncarriers (p<0.03). Specifically, they found evidence of a significant incidence of cervical stenosis in carriers 5/34 carriers compared to 0/31 non-carriers (Fisher exact test p<0.05). This was especially interesting as cervical stenosis, or narrowing of the upper spinal canal, is thought to be rare in individuals under the age of 50. Although not reaching statistically significance, they also found a trend of 2-fold higher rates of scoliosis (24%; p<0.35), arthritis (38%; p<0.1), and lumbar spine degradation (29%; p<0.25) in carriers compared to non-carriers and previous published reports. We report the new analysis of the detailed chart review and additional PheWAS models in the Results, Discussion, Table 1, Figure 3—figure supplement 1 and Materials and methods.

3) Power of IBD mapping – First, currently no power calculations are presented to substantiate the claim that the IBD mapping approach outperforms GWAS in terms of power. Please present these calculations as the p-value yielded through 10,000 permutations is only p=0.0018, which is substantially less than a GWAS significance threshold. Second, please clarify if the COL27A1.pG697R variant is imputable with 1000 Genomes reference. In the Materials and methods section, you provide detailed description on imputation. Furthermore in paragraph two of subsection “Fine-mapping Short Stature Locus Reveals Putative Link to Mendelian Syndrome” it states that there was 96.7% concordance between variant genotype and the associated haplotype – so wouldn't the GWAS have identified the association using the variant?

The specific issue here is that IBD tagged the three true homozygotes perfectly, whereas a standard imputation pipeline (SHAPEIT2/IMPUTE2/1000 Genomes Project phase 3 reference panel) called one of the homozygotes as a heterozygote. Running the same recessive model in both cases results in an difference of the p-value of association from p<2.57x10-11 (number of homozygotes=3; IBD-clique frequency=0.012; β=- 3.78; rank=1) to p<0.001 (number of homozygotes=2; MAF=0.014: β=-3.0; rank=11775) comparing IBD mapping with SNP imputation, respectively. In addition, there were over 10 million SNP tests performed in the imputed dataset compared to the 480 IBD-cliques tested in the IBD mapping approach, so the Bonferonni adjusted genome-wide significance testing burden is considerably reduced in the IBD mapping approach. Despite the frequencies and effects sizes being similar for both approaches, the p-values and absolute rank of the test of association are diminished for the imputation approach. Thus, in this case, IBD mapping was better powered compared to the SNP imputation approach for association of the recessive COL27A1.pG697R allele. Recent methods that leverage IBD for imputation, such as EAGLE (Loh et al.,Nat Genet 2016), have demonstrated improved accuracy of low frequency and rare variant imputation and may have improved imputation of the homozygotes in this study. However, the broader point is that imputation methods are dependent on representative haplotypic diversity in the sequence reference panels. Since rare disease variants are more likely to be population-specific or geographically localized, lacking the relevant diversity in sequence reference panels or sequence discovery efforts means that inference of rare variants within cohorts independent of available reference panels via methods like IBD mapping, which rely on inferring distant familial relationships, are likely to remain important for disease variant discovery.

We have updated the text to clarify details of this analysis in subsection “Fine-mapping Short Stature Locus Reveals Putative Link to Mendelian Syndrome”, Figure 2—figure supplement 4, Supplementary file 3.

4) Genetic ancestry and COL27A1.pG697R variant – Currently the text is lacking detailed description of the pathogenic variant. Furthermore, more detail is essential to understand how partitioning into ancestry groups was performed.First, in the functional analysis section (subsection “Functional Investigation of COL27A1.pG697R”). Is the variant predicted damaging by popular annotation tools?- polyphen, sift, cadd, gerp, mutation taster, phyloP etc.Second, please provide a confidence interval for the dating of the variant, (around 36 generations).Third, please provide exact criteria how ancestry groups were defined from PCA vectors. Such details are currently missing from the Materials and methods. Furthermore, please provide clear evidence that suggest the populations that are described in subsection “Local Ancestry Estimation” present Native American are not admixed for European ancestry, or at least not significantly so.

We have added details to the text to clarify the first and third points in the Results and Materials and methods sections, Figure 1—figure supplement 4 and Figure 4—figure supplement 2. For the second point, we cannot provide a confidence interval as the estimate was based on two heterozygote observations (in the 100 sequenced Puerto Rican’s in the 1000 Genomes Project). Since we have no way to infer the stability of the estimate, we removed this analysis from the paper. Instead we add a new analysis to estimate the timing and magnitude of the putative founder event in Puerto Rico by demographic modeling using the tract length distribution of IBD tracts in contemporary Puerto Rican populations with IBDNe (Browning and Browning, Am J Hum Genet 2015). We estimate a strong bottleneck in Puerto Rico dating approximately 12 generations ago (estimated effective population size=2580, 95% C.I 2320-2910). This is described in the Results, Discussion and Materials and methods sections, and Figure 4.